# Wukong's 72 Transformations: High-fidelity Textured 3D Morphing via Flow Models

**Minghao Yin**[1]     **Yukang Cao**[2]     **Kai Han**[1]*

[1]Visual AI Lab, The University of Hong Kong     [2]S-Lab, Nanyang Technological University

`yinmh@connect.hku.hk`     `yukang.cao@ntu.edu.sg`     `kaihanx@hku.hk`

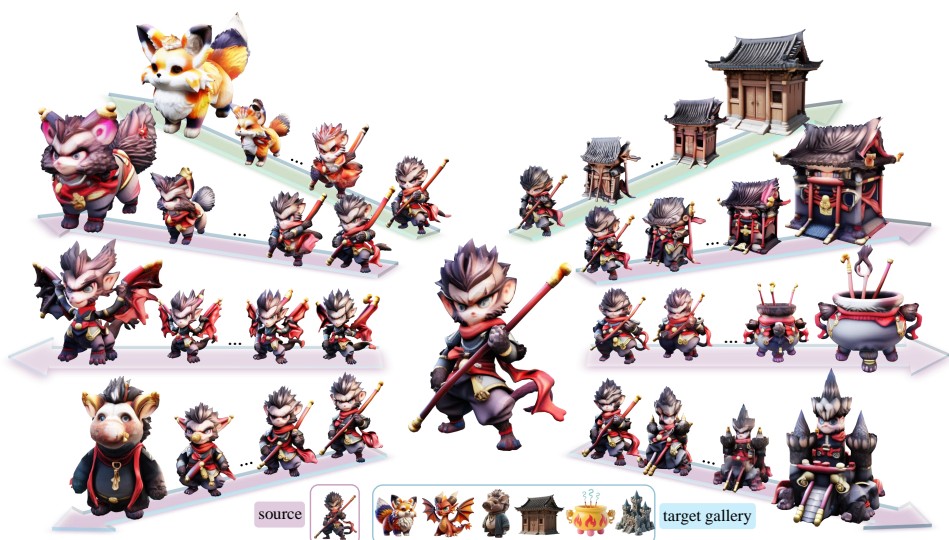

Figure 1: **High-fidelity textured 3D morphing of Wukong.** Taking an image of Wukong (bottom left) as the source and an image of another character (bottom right) as the target, we demonstrate two types of *textured 3D morphing* by our method: (i) Purple arrows indicate texture-controlled morphing, where the geometric structure changes while preserving detailed textures from the source; (ii) Green arrows indicate textured 3D morphing guided by the target prompt.

## Abstract

We present WUKONG, a novel training-free framework for high-fidelity textured 3D morphing that takes a pair of source and target prompts (image or text) as input. Unlike conventional methods—which rely on manual correspondence matching and deformation trajectory estimation (limiting generalization and requiring costly preprocessing)—WUKONG leverages the generative prior of flow-based transformers to produce high-fidelity 3D transitions with rich texture details. To ensure smooth shape transitions, we exploit the inherent continuity of flow-based generative processes and formulate morphing as an optimal transport barycenter problem. We further introduce a sequential initialization strategy to prevent abrupt geometric distortions and preserve identity coherence. For faithful texture preservation, we propose a similarity-guided semantic consistency mechanism that selectively retains high-frequency details and enables precise control over blending dynamics. This empowers WUKONG to support both global texture transitions and identity-preserving texture morphing, catering to diverse generation needs. Extensive quantitative and qualitative evaluations demonstrate that WUKONG significantly outper-

---

*Corresponding author.

39th Conference on Neural Information Processing Systems (NeurIPS 2025).

forms state-of-the-art methods, achieving superior results across diverse geometry and texture variations. Project page: `https://visual-ai.github.io/wukong`

## 1 Introduction

3D morphing techniques (Seitz and Dyer, 1996; Shechtman et al., 2010; Tsai et al., 2022; Kim et al., 2024) aim to produce smooth transitions between the source object and target object by gradually altering 3D attributes such as shape and texture. These methods have broad applications in gaming, animation, and cinematic transitions. Existing approaches primarily focus on geometric transformations, relying on correspondence matching (Deng et al., 2023) and deformation trajectory estimation (Eisenberger et al., 2021). However, most existing methods are limited to untextured 3D meshes, leaving textured 3D morphing a largely under-explored problem. Moreover, their effectiveness is constrained by the limited availability of large-scale 3D datasets with source-target pairs, often resulting in unsatisfactory outputs when applied to unseen data. To bridge this gap, we introduce WUKONG, a method for *high-fidelity textured 3D morphing from a pair of image or text prompts*. This name is inspired by the legendary character Wukong and his 72 earthly transformations, as exemplified by the morphing result in Fig. 1.

With recent advances in large-scale 3D generation and reconstruction (Xiang et al., 2024; Tochilkin et al., 2024; Li et al., 2025), one may consider to address the 3D morphing problem by training a feed-forward network to model the transitions between two input objects. However, constructing paired 3D data for training based on existing large-scale 3D data (Deitke et al., 2024) is non-trivial, both technically and economically. Therefore, such a straightforward approach is infeasible.

Recently, 2D image morphing has achieved remarkable success (Qiyuan et al., 2024; Zhang et al., 2024a), driven by advances in 2D diffusion models (*e.g.*, Stable Diffusion (Rombach et al., 2022)). Inspired by this progress, we aim to extend these successes to 3D – morphing not only shapes but also textures – without relying on large-scale paired data. In other words, we aim to develop a training-free framework for textured 3D morphing by leveraging the strong priors of generative models. However, building such a framework poses significant challenges due to the absence of large-scale 3D data with continuous shape and texture transitions.

In this paper, we propose WUKONG, a novel framework for high-fidelity textured 3D morphing that takes image or text pairs as input to delineate source and target objects. Built upon a pre-trained flow-based transformer (Xiang et al., 2024) for 3D generation, WUKONG bridges the gap between 3D shapes and image/text conditions. Leveraging the deterministic property of flow models (where intermediate states are not stochastic), we derive morphing trajectories by solving a free-support Wasserstein barycenter problem. Additionally, we introduce a sequential initialization strategy to enhance the smoothness of the transitions along the barycentric trajectory. This design also allows us to handle the texture morphing in 3D in a similar fashion. However, a naive interpolation treats all texture regions uniformly, which may not satisfy the need for preserving specific source attributes, such as fine details, distinct styles, or identity patterns. To address this, we propose a similarity-guided consistency mechanism for texture controlled morphing, which selectively preserves high-frequency texture details while providing finer control over transition dynamics.

The main contributions of this work are as follows: (i) We propose WUKONG, a novel framework for high-fidelity textured 3D morphing that takes a pair of image or text prompts as input. By leveraging a flow-based generative model as a prior, WUKONG enables smooth and controllable shape and texture interpolation. (ii) We introduce a method to derive faithful intermediate morphing states by solving an optimal transport barycenter problem, further augmented by a sequential initialization strategy to facilitate smooth transitions. (iii) We propose a similarity-guided semantic consistency mechanism to enable texture controlled morphing, allowing users to selectively preserve high-frequency texture details and exercise fine-grained control over appearance transitions. (iv) Through extensive experiments on diverse 3D morphing scenarios—spanning different object categories with varying geometry and significant texture changes—we demonstrate that WUKONG outperforms existing methods, establishing a new state-of-the-art in textured 3D morphing.

## 2 Related work

**2D morphing**  Classical image morphing (Liao et al., 2014; Beier and Neely, 1992; Darabi et al., 2012; Shechtman et al., 2010) typically involves three key steps: (1) finding feature-based correspondence; (2) mapping between two images using optimization frameworks; and (3) auxiliary techniques

to ensure smooth transitional continuity. However, strong priors and limited model expressiveness often lead to ghosting artifacts. With growing data availability, data-driven morphing methods (Fish et al., 2020; Averbuch-Elor et al., 2016) shifted from explicit mappings to learning over dataset distributions using deep models. However, their generalization is limited by dataset-specific training. Recent approaches like DiffMorpher (Zhang et al., 2024a) address this by leveraging pre-trained diffusion models, enabling more flexible morphing. DiffMorpher achieves morphing by interpolating noise, conditional inputs, and selectively blending model parameters, enabling strong shape and texture transitions. Inspired by this idea, we propose a novel method that replaces linear interpolation with an optimal transport-based strategy, offering a more principled and effective interpolation of latent conditions.

**3D generation**  To achieve 3D generation, two dominant paradigms have emerged: (1) Distilling useful priors from pretrained generative models into a 3D feed-forward reconstruction algorithm; (2) Training a unified 3D generative model from scratch. In the first paradigm, knowledge distillation from large generative models occurs via gradient-based or data-based methods. Data distillation fine-tunes 2D models to generate multi-view images (Yu et al., 2024; Shi et al., 2024; Shriram et al., 2025), which are then reconstructed into 3D using methods like Gaussian splatting (Kerbl et al., 2023). Gradient distillation, exemplified by Score Distillation Sampling (Poole et al., 2023), guides 3D optimization directly. However, both lack a latent 3D space, limiting structural control. To bridge this gap, native 3D generative models have emerged (Lan et al., 2025; Zeng et al., 2022; Zhang et al., 2024b), typically combining a VAE for dimensionality reduction with a 3D generative model. However, most are limited to either explicit (*e.g.*, point clouds, voxels, meshes) or implicit (*e.g.*, neural fields, Gaussians) formats. Trellis (Xiang et al., 2024) addresses this by introducing a Structured Latent Representation (SLAT) for flexible multi-format generation. We inherit Trellis's versatility to support diverse 3D modalities.

**3D morphing**  Early 3D morphing research centered on shape transition (Tam et al., 2013), following a three-step pipeline: finding correspondence, modeling the mapping, and refinement. Classical methods employed techniques like Wasserstein distance (Tsai et al., 2022; Solomon et al., 2015; Sorkine and Alexa, 2007; Ren et al., 2020; Eisenberger et al., 2021). Despite their contributions, these methods suffer from oversimplified assumptions and high input sensitivity. Recent learning-based methods improve robustness by integrating foundation model priors. NSSM (Morreale et al., 2024) uses DINOv2 (Oquab et al., 2023) for 3D correspondence; SRIF (Sun et al., 2024) incorporates a diffusion prior from DiffMorpher (Zhang et al., 2024a). Emerging works (Gao et al., 2023; Yang et al., 2025; Michel et al., 2022) explore text-driven morphing using CLIP (Radford et al., 2021), though limited by CLIP's coarse latent space. Other efforts leverage topology-aligned datasets (Eisenberger et al., 2021) with in-domain training to enhance semantic consistency (Yumer et al., 2015). Shape-only 3D morphing has limited practical use without texture and appearance cues. A recent concurrent work (Yang et al., 2025) explores morphing textured 3D representations using generative models, bypassing explicit correspondence computation. However, it mainly focuses on parameter fusion and pays less attention to latent condition interpolation. In contrast, our method emphasizes a principled interpolation schedule for latent conditions. Additionally, we adopt a flow-based model with a unified 3D representation, enabling flexible output across diverse formats.

## 3  Method

### 3.1  Overview and formulation

Given two input prompts $(P_{\text{source}}, P_{\text{target}})$ in image or text form, our method learns an interpolation function $\mathbf{I}$ to generate a coherent sequence of 3D textured meshes $\mathcal{G} = \{\mathbf{G}_\alpha\}_{\alpha=0}^{J+1}$, which forms a smooth morphing trajectory between the two concepts. Here, $\alpha \in \{0, \ldots, J+1\}$ denotes the discrete interpolation step index. Among them, $\mathbf{G}_0$ and $\mathbf{G}_{J+1}$ denote the 3D generation of the input prompts $P_{\text{source}}$ and $P_{\text{target}}$, respectively. Formally, the problem can be formulated as:

$$\mathbf{G}_\alpha = \Phi(\mathbf{I}(\mathcal{E}(P_{\text{source}}), \mathcal{E}(P_{\text{target}}), \alpha)), \tag{1}$$

where $\Phi$ denotes a flow-based 3D generator, $\mathcal{E}$ represents the encoders that embed the prompts into the latent space. For feature extraction, we use DINOv2 (Oquab et al., 2023) as the encoder $\mathcal{E}$ for image prompts and CLIP (Radford et al., 2021) for textual inputs.

In the rest of this section, we will discuss the details of our employed flow-based 3D generator $\Phi$ and how we formulate the interpolation function $\mathbf{I}$. From a general perspective, an effective interpolation

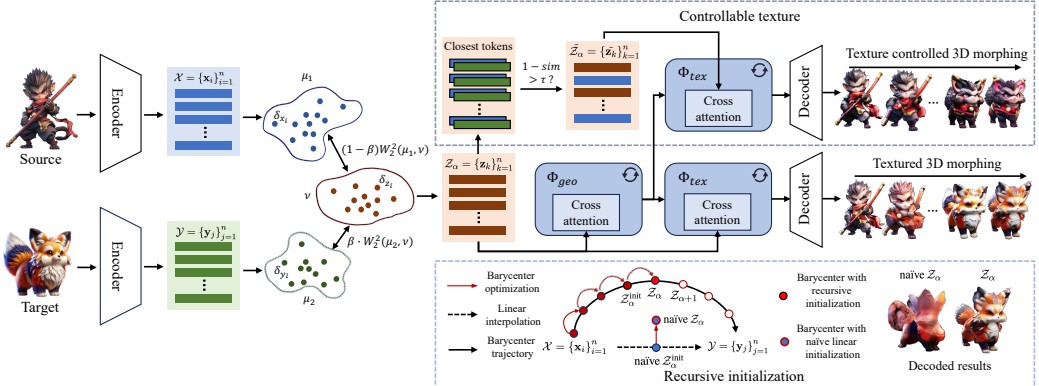

Figure 2: **Our WUKONG framework.** Given a source and a target (image or text), we extract features using pretrained encoders and treat the condition tokens as empirical distributions. We compute their Wasserstein barycenter with weights $(1 - \beta)$ and $\beta$ to obtain interpolated condition tokens $\mathcal{Z}$. These tokens are fed into a shared geometry flow model $\Phi_{geo}$ and texture flow model $\Phi_{tex}$ to generate 3D outputs at different $\alpha$ values, producing textured 3D morphs. The top-right shows our texture-controlled morphing branch, and the bottom-right illustrates the recursive initialization strategy.

strategy in 3D morphing should satisfy two key requirements: (1) smooth shape transitions that preserve identity and prevent abrupt changes, and (2) controllable texture blending. As depicted in Fig. 2, we propose to model these two properties with (1) shape interpolation via optimal transport barycenter and (2) controllable texture generation via selective interpolation.

## 3.2 Flow-based 3D generator

Recently, Trellis (Xiang et al., 2024) advances 3D generation by introducing a unified structured latent representation via rectified flow transformers (Xiang et al., 2024). It achieves high-fidelity outputs from image or text inputs by training a feed-forward pipeline, with the generation process at test time separated into three distinct phases:

*(1) Geometric structure generation* that generates sparse structure $\boldsymbol{p} = \{p_i\}_{i=1}^{L}$ from the condition $\mathcal{C}$, which are either text-embedded CLIP features (Radford et al., 2021) or image-encoded DINOv2 features (Oquab et al., 2023):

$$\{p_i\}_{i=1}^{L} = \Phi_{\text{geo}}(\mathcal{C}, t), \tag{2}$$

where $\Phi_{\text{geo}}$ denotes the flow transformer backbone, $t$ is the timestep, and $L$ represents the number of active voxels.

*(2) Textured latent generation* that generates latents $\boldsymbol{h} = \{h_i\}_{i=1}^{L}$ given the structure $\{p_i\}_{i=1}^{L}$:

$$\{h_i\}_{i=1}^{L} = \Phi_{\text{tex}}(\boldsymbol{p}, \gamma(\boldsymbol{x}), \mathcal{C}, t), \tag{3}$$

where $\gamma(\boldsymbol{x})$ is the positional embedded points.

*(3) Latent decoder* that decodes the structured latents $\boldsymbol{s} = \{(h_i, p_i)\}_{i=1}^{L}$ into a 3D representation, which we opt for meshes in our experiments:

$$\mathcal{D}_{\text{mesh}}(\boldsymbol{s}) \rightarrow \{\{w_i^j, d_i^j\}_{j=1}^{64}\}_{i=1}^{L}, \tag{4}$$

where $w_i^j \in \mathbb{R}^{45}$ are the flexible parameters in FlexiCubes (Shen et al., 2023) and $d_i^j \in \mathbb{R}^8$ are signed distance values for the eight vertices of the corresponding voxel. Note that our employed 3D generator (Xiang et al., 2024) also supports output in both 3D Gaussian Splatting (Kerbl et al., 2023) and NeRF (Martin-Brualla et al., 2021) formats, and our method naturally inherits this capability.

For our experiment, we utilize the pre-trained Trellis model as the backbone network due to its demonstrated efficiency and high-quality outputs:

$$\Phi = \{\Phi_{\text{geo}}, \Phi_{\text{tex}}, \mathcal{D}_{\text{mesh}}\}. \tag{5}$$

To maintain distinct control over different attributes, we design separate interpolation functions for geometric and texture features:

$$\mathbf{I} = \{\mathbf{I}_{\text{geo}}, \mathbf{I}_{\text{tex}}\}. \tag{6}$$

### 3.3 Shape interpolation via optimal transport barycenter

Given $\mathcal{C}_0 = \mathcal{E}(P_{\text{source}})$ and $\mathcal{C}_{J+1} = \mathcal{E}(P_{\text{target}})$, which represent the conditioning features respectively encoded from the input prompts $P_{\text{source}}$ and $P_{\text{target}}$, we first model them as discrete probability distributions in feature space by extracting their respective sets of feature tokens:

$$\mathcal{X} = \{\mathbf{x}_i\}_{i=1}^n \subset \mathbb{R}^m, \quad \mathcal{Y} = \{\mathbf{y}_j\}_{j=1}^n \subset \mathbb{R}^m, \tag{7}$$

where $m$ is the embedding dimension. We can then obtain their empirical distribution:

$$\mu_1 = \sum_{i=1}^n a_i \delta_{\mathbf{x}_i}, \quad \mu_2 = \sum_{j=1}^n b_j \delta_{\mathbf{y}_j}, \tag{8}$$

where $\delta_{\mathbf{x}_i}$ and $\delta_{\mathbf{y}_j}$ denote the Dirac measures located at the tokens $\mathbf{x}_i$ and $\mathbf{y}_j$, respectively. $a_i$ and $b_j$ are weights assigned to each token, which we set $a_i = b_j = 1/n$ in our experiments.

To obtain the geometric structure $\boldsymbol{p}_\alpha$ at an intermediate step $\alpha$ via the generator $\Phi_{\text{geo}}$, we require its corresponding interpolated condition $\mathcal{Z}_\alpha$. Since $\Phi_{\text{geo}}$ is a deterministic mapping from conditions to structures, we propose to obtain $\mathcal{Z}_\alpha$ by solving a free-support Wasserstein barycenter problem:

$$\mathbf{I}_{\text{geo}} : \mathcal{Z}_\alpha = \arg\min_{\mathcal{Z}}[(1-\beta)\cdot\mathcal{W}_2^2(\mu_1,\nu) + \beta\cdot\mathcal{W}_2^2(\mu_2,\nu)], \quad \beta = \alpha/(J+1),$$

$$\mathcal{W}_2^2(\mu,\nu) = \inf_{\gamma\in\Gamma(\mu,\nu)} \int_{n\times n} d(x,y)^2\,\mathrm{d}\gamma(x,y), \tag{9}$$

where $\mathcal{W}_2^2(\mu,\nu)$ denotes the 2-Wasserstein distance, $\nu = \sum_{k=1}^n c_k \delta_{\mathbf{z}_k}$ is the target interpolated distribution with learnable support points $\{\mathbf{z}_k\}_{k=1}^n$ uniformly weighted by $c_k = 1/n$. The interpolated tokens $\mathcal{Z}_\alpha = \{\mathbf{z}_k\}_{k=1}^n$ generated through this process serve as input to our geometric structure generator $\Phi_{\text{geo}}$, which produces the corresponding 3D structure $\boldsymbol{p}_\alpha$.

A critical design for solving the free-support Wasserstein barycenter problem is the initialization of support points $\{\mathbf{z}_k\}$. While a naïve linear interpolation approach, $\mathbf{z}_{k,\text{init}} = (1-\beta)\cdot\mathbf{x}_k + \beta\cdot\mathbf{y}_k$, might seem reasonable, we empirically observe that this often results in discontinuous or inconsistent interpolations. This is particularly evident when interpolating between either: (1) geometrically distant shapes, or (2) semantically divergent conditions (see Fig. 8). The failure arises because linear initialization overlooks the structure of token distributions, often leading the barycenter optimization to unstable or unrealistic interpolations.

We address this limitation with a sequential initialization scheme that guarantees smooth transitions along the barycenter trajectory. Specifically, for interpolation step $\alpha \in [0, J+1]$, we initialize its barycenter using the optimized solution from the previous step:

$$\mathbf{z}_{k,\text{init}}^{(\alpha)} = \begin{cases} \mathbf{x}_k, & \text{if } \alpha = 0 \\ \mathbf{z}_k^{(\alpha-1)}, & \text{otherwise}. \end{cases} \tag{10}$$

This recursive approach supports the barycenter evolves continuously on the data manifold, maintaining: (1) geometric coherence: support points adapt gradually, reducing abrupt deviations or poor local minima, and (2) semantic stability: intermediate shapes retain recognizable object parts and consistent structural semantics throughout the interpolation.

### 3.4 Controllable texture generation via selective interpolation

As illustrated in Fig. 2, our framework supports two strategies for texture evolution. The default textured 3D morphing utilizes the barycenter interpolated tokens $\mathcal{Z}_\alpha$ (derived in Sec. 3.3) as the condition for the texture flow model $\Phi_{tex}$. Although this produces a smooth transformation, it strictly couples texture evolution with the structural transition. However, creative applications often demand more flexibility—specifically, the ability to preserve the source's visual identity (e.g., facial features or iconic patterns) even as the shape transforms. One may consider enforcing source tokens $\mathcal{X}$ across all token regions, but this is not feasible: as the geometry deforms, forcing static tokens onto the evolving structure creates inherent texture-geometry misalignment, resulting in visual artifacts.

To address this, we introduce texture controlled 3D morphing. This mode employs a selective interpolation strategy that functions as a tunable filter. By adjusting a similarity threshold, users

can explicitly regulate the degree of source preservation, spanning from subtle detail retention to strong identity maintenance. This mechanism retains high-frequency details from $\mathcal{X}$ only where semantically aligned to prevent artifacts, while utilizing $\mathcal{Z}_\alpha$ elsewhere to ensure structural coherence.

Building upon the geometric interpolation introduced in Sec. 3.3, we first compute the free-support barycenter $\mathcal{Z}_\alpha = \{\mathbf{z}_k\}_{k=1}^n$ from Eq. 9. While this provides a general interpolation baseline, it may dilute fine-grained details from the source. To mitigate this, we augment the interpolation with a semantic consistency evaluation between barycenter points $\mathbf{z}_k$ and the source tokens $\mathcal{X}, \mathcal{Y}$.

Specifically, for each barycenter point $\mathbf{z}_k$, we identify its closest source tokens $\mathbf{x}_i$ and $\mathbf{y}_j$ and compute the cosine similarity $\mathtt{sim} = \cos(\mathbf{x}_i, \mathbf{y}_j)$. We then selectively retain high-frequency information from either $\mathbf{x}_i$ or the interpolated tokens based on similarity:

$$\mathbf{I}_{\text{tex}} : \tilde{\mathcal{Z}}_\alpha = \{\tilde{\mathbf{z}}_k\}, \quad \text{while} \quad \begin{cases} \tilde{\mathbf{z}}_k = \mathbf{z}_k, & \text{if } 1 - \mathtt{sim} > \tau \\ \tilde{\mathbf{z}}_k = \mathbf{x}_i, & \text{otherwise.} \end{cases} \tag{11}$$

Here, $\tau \in [0, 1]$ is a pre-defined similarity threshold that determines whether semantic discrepancy is large enough to justify interpolation. See Fig. 6 for an analysis across different values of $\tau$. This approach enables asymmetric texture fusion, where more visually salient or personalized texture features can be preserved from one condition, while maintaining semantically meaningful global structure via the barycenter.

The selectively refined condition tokens $\tilde{\mathcal{Z}}_\alpha$ are then passed into the 3D textured structured latents generator $\Phi_{\text{tex}}$ to produce the final latent representation $\boldsymbol{h}_\alpha$. The interpolated 3D textured mesh at step $\alpha$ can then be obtained through the decoding function Eq. 4:

$$\mathbf{G}_\alpha = \mathcal{D}_{\text{mesh}}(\boldsymbol{s}_\alpha), \quad \text{where } \boldsymbol{s}_\alpha = \{\boldsymbol{h}_\alpha, \boldsymbol{p}_\alpha\} = \{(h_{i,\alpha}, p_{i,\alpha})\}_{i=1}^L. \tag{12}$$

## 4 Experimental results

**Implementation details** We adopt the Trellis framework as our 3D generative model. Both the structure flow and texture flow models are implemented using rectified flow with 25 sampling steps each. The Classifier-Free Guidance (CFG) strength is set to 3. DINOv2 (Oquab et al., 2023) and CLIP (Radford et al., 2021) are used for image and text feature extraction. Both the structure and texture flow-based generator $\Phi_{\text{geo}}$, $\Phi_{\text{tex}}$ contain 21 cross-attention blocks, where interpolation is performed on every condition token before each cross-attention layer. The morphing coefficient $\alpha$ is uniformly sampled during morphing and we set $J = 6$ for experiments presented in the paper. For shape interpolation, we compute the token-wise barycenter using the free-support Wasserstein barycenter implemented by `ot.lp.free_support_barycenter` (Lindheim, 2023). We set the maximum number of optimization iterations to 100, and the convergence threshold (stop criterion) to $1 \times 10^{-5}$. We conduct experiments on an NVIDIA A100 GPU. Generating a single morphed 3D output takes approximately 30 seconds.

**Metrics** Following (Yang et al., 2025), we evaluate textured 3D morphing quality using metrics for fidelity, plausibility, and smoothness on their input pairs: (1) FID (Heusel et al., 2017) for visual fidelity; (2) STP-GPT and SEP-GPT for structural and semantic consistency; (3) GPT-4o (Hurst et al., 2024) for visual plausibility (0–1 score); (4) PPL (Karras et al., 2019) for perceptual smoothness; and (5) V-CLIP (Ma et al., 2022), which measures semantic alignment to "a smooth transformation from A to B" using cosine similarity in a joint embedding space.

**Baseline methods for evaluation** We compare our 3D textured morphing results against two baseline methods: 3DRM (Yang et al., 2025) and MorphFlow (Tsai et al., 2022). To further evaluate performance, we render the 3D meshes into 2D images and compare them with state-of-the-art 2D image morphing approaches, including DiffMorpher (Zhang et al., 2024a), AID (Qiyuan et al., 2024), MV-Adapter (Jin et al., 2024), and Luma (Luma Labs AI, 2025).

### 4.1 Main results

#### 4.1.1 Quantitative results

In Tab 1, we compare our method against a range of baselines across multiple evaluation metrics. As can be observed, our method consistently outperforms existing approaches across all metrics,

Table 1: **Quantitative comparison.**

| Model | FID ↓ | STP-GPT ↑ | SEP-GPT ↑ | PPL ↓ | V-CLIP ↑ |
|---|---|---|---|---|---|
| DiffMorpher | 218.07 | 0.14 | 0.10 | 5.23 | 0.61 |
| AID | 115.72 | 0.46 | 0.62 | 4.68 | 0.74 |
| MV-Adapter | 120.93 | 0.44 | 0.49 | 7.29 | 0.67 |
| Luma | 95.49 | 0.69 | 0.65 | 7.37 | 0.70 |
| MorphFlow | 147.70 | 0.71 | 0.79 | 3.10 | 0.78 |
| 3DRM | 6.36 | 0.93 | 0.88 | 3.02 | 0.84 |
| **Ours** | **4.01** | **1.00** | **1.00** | **2.91** | **0.90** |

demonstrating superior quality and consistency in both shape and texture morphing. Specifically, we outperform state-of-the-art textured 3D morphing methods across the board, *i.e.*, 3DRM (Yang et al., 2025) and MorphFlow (Tsai et al., 2022), showing clear improvements in perceptual quality and semantic coherence. Furthermore, compared with image-based morphing methods (Zhang et al., 2024a; Qiyuan et al., 2024; Jin et al., 2024; Luma Labs AI, 2025), our approach also exhibits much stronger 3D consistency and higher fidelity in both appearance and structure. See Appendix B for more quantitative evaluations.

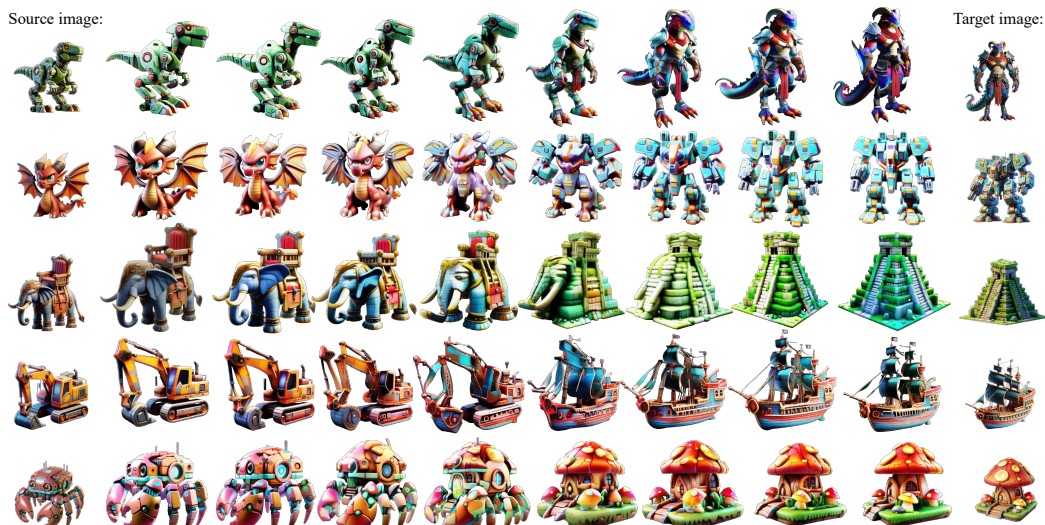

Figure 3: **Textured 3D morphing with image prompts.** The leftmost column shows source image prompts, while the rightmost column shows target prompts. Intermediate columns depict the morphing trajectory generated by our method.

### 4.1.2 Qualitative results

**Image-conditioned 3D morphing** Fig. 3 shows image-conditioned 3D morphing results. Each row presents a smooth transition from source to target, with consistent shape and texture interpolation. Intermediate frames preserve structure without distortions. Rows 3–5 highlight our method's ability to handle cross-category morphing with clear semantic consistency.

**Text-conditioned 3D morphing** Our method also enables 3D morphing conditioned on textual descriptions, allowing users to generate transitions directly from text prompts. As shown in Fig. 4, the results exhibit smooth transitions in both shape and texture, with intermediate outputs maintaining high fidelity and semantic alignment. Notably, the second row captures a precise castle-to-room transformation, while the fourth row demonstrates realistic face morphing with consistent structure.

**Comparison with existing methods** We now compare our method with the current state-of-the-art in textured 3D morphing (3DRM (Yang et al., 2025)) in Fig. 5. Visually, our method delivers higher overall quality with significantly smoother transitions in both geometry and texture. Color variations in our morphing sequences also appear more continuous, and the interpolated shapes retain clear

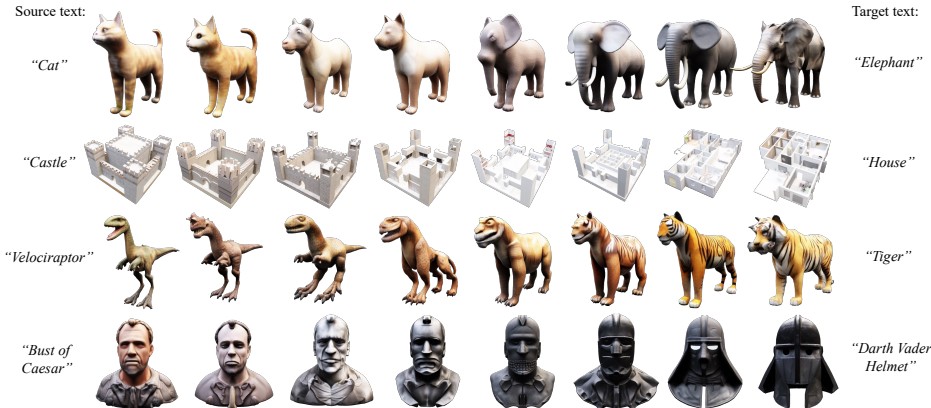

Figure 4: **Textured 3D morphing with text prompts.** The leftmost column shows the source text descriptions, and the rightmost column shows the target prompts. Intermediate results visualize the smooth morphing trajectory generated by our method.

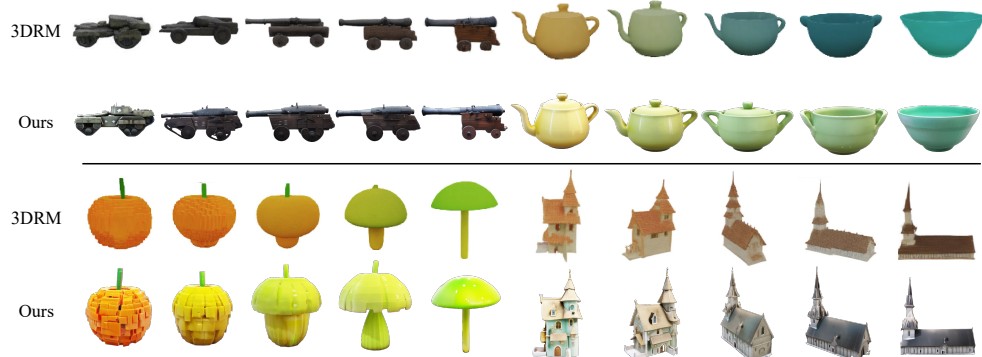

Figure 5: **Qualitative comparison with current SOTA method 3DRM.** Rows 1 and 3 show 3DRM's generated results, while rows 2 and 4 display our method's outputs.

semantic coherence throughout. We further compare with MorphFlow, a recent textured 3D morphing approach, in the Appendix D.

**Controllable texture morphing** In Fig. 6, we demonstrate the capabilities of our texture controlled morphing mode, by employing our proposed *selective texture interpolation*. Each row represents a distinct morphing process with different interpolation thresholds, with the leftmost column fixed as the source shape. The results along the vertical axis (columns) are independent of each other. The far-left column does not show texture changes, as it represents the same 3D source generated from the input source image. By adjusting the selective interpolation threshold $\tau$ (from top to bottom), we can control the relative influence of source and target tokens during the morphing process. For instance, we can preserve source-dominant features such as facial identity, clothing styles, and color patterns in the intermediate 3D shapes, while still achieving smooth and semantically coherent shape transitions.

## 4.2 Ablation study

**Ablation on shape interpolation** In Fig. 7 (top three rows), we first present ablation studies evaluating different strategies for shape interpolation in image-conditioned 3D morphing. Specifically, (1) the first row shows results from directly applying linear interpolation to the condition tokens. While this produces smooth blending in token space, it results in semantically ambiguous intermediate shapes, incoherent textures, and noticeable color artifacts, revealing the limitations of naïve token-level averaging. (2) The second row presents our method using a linearly initialized free-support barycenter (i.e., linear initialization of support points). Although more flexible than fixed linear interpolation, it tends to converge to undesirable solutions, leading to distorted geometry and unstable textures in intermediate outputs. (3) In contrast, the third row shows our full method, where the

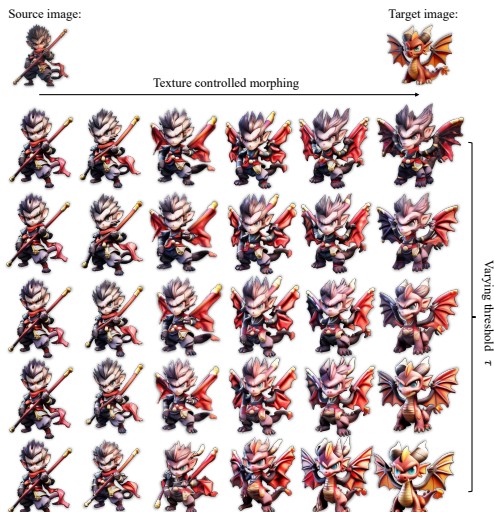

Figure 6: **Texture controlled morphing results.** The figure shows shape interpolation arranged horizontally, with each row representing a morphing process at different interpolation thresholds, illustrating the smooth transition from the source image to the target image.

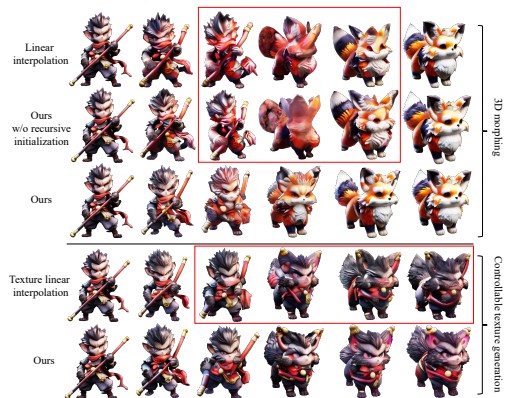

Figure 7: **Image-conditioned ablation study.** Top three rows compare textured 3D morphing: linear interpolation and no recursive initialization show shape and color artifacts, while ours is clean. Bottom two rows compare texture control: increasing source weight preserves details but distorts color; our selective interpolation maintains texture quality.

barycenter is properly initialized and refined through iterative optimization. This results in high-quality morphs with semantically meaningful structure, smooth geometric transitions, and coherent color blending.

We further conduct ablation studies for shape interpolation under the setting of text-conditioned 3D morphing, as in Fig. 8. The results reveal the following: (1) Linear interpolation (row 1) produces unnatural artifacts, such as T-pose human figures, indicating that direct token blending often strays into semantically invalid regions. (2) Our method without recursive initialization (row 2) exhibits similar issues, underscoring the importance of proper initialization for barycenter optimization. Linear initialization fails to respect the underlying data manifold, often leading to unstable or artifact-prone results, as detailed in Sec. 3.3. (3) Our full method (row 3), with recursive initialization, successfully avoids these problems, delivering smooth and structurally plausible shape transitions throughout the morph.

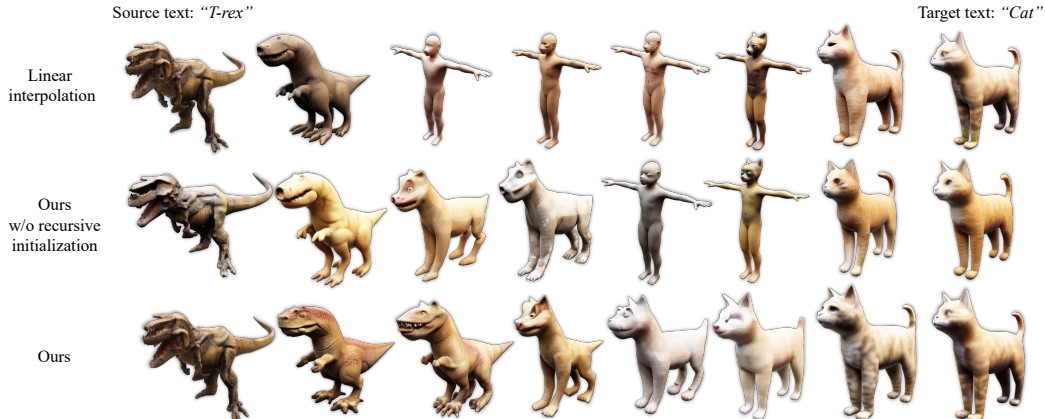

Figure 8: **Ablation study on text-conditioned 3D shape morphing.** Morphing from "T-rex" to "cat," the first row (linear interpolation) shows multiple T-pose human artifacts. The second row (without recursive initialization) still has some artifacts. The third row (full method) achieves smooth, coherent shape transitions.

In flow-based 3D generators, condition tokens guide the generation by modulating cross-attention at each layer. Linear interpolation between latent features explores intermediate regions of the latent

space. However, when the source and target features are semantically distant (*e.g.*, from "T-Rex" to "cat" in Fig. 8) and lack appropriate supervision, the model's conditioning may become ambiguous or collapse into mode-averaged representations. This often results in "hallucinated" generic outputs, such as T-poses or default humanoid templates commonly seen in generative models. This issue highlights the importance of our barycentric interpolation and sequential initialization strategy, which helps: (1) intermediate tokens remain on a semantically valid manifold; (2) abrupt jumps across unrelated modes are mitigated.

For quantitative comparison, Tab 2 provides ablation results to validate our method. The "Linear" baseline, using direct interpolation without optimization, leads to unnatural shapes and visible artifacts. The "w/o r-i" row employs optimization but omits recursive initialization; while it improves upon the linear baseline, the lack of a robust prior often causes convergence to suboptimal solutions, resulting in distortions. In

Table 2: **Ablation on shape interpolation.**

| Model | FID ↓ | STP-GPT ↑ | SEP-GPT ↑ | PPL ↓ | V-CLIP ↑ |
|---|---|---|---|---|---|
| Linear | 14.73 | 0.86 | 0.85 | 3.06 | 0.81 |
| w/o r-i | 12.52 | 0.90 | 0.92 | 3.03 | 0.84 |
| Ours* | 4.16 | 0.96 | **0.98** | 2.95 | **0.91** |
| **Ours** | **4.01** | **0.97** | 0.95 | **2.91** | 0.90 |

contrast, "Ours*" applies our full method to the TripoSG (Li et al., 2025) backbone. This configuration includes recursive initialization, which acts as a strong prior to guide optimization toward stable and meaningful representations. Consequently, "Ours*" yields smoother geometry and better texture blending than the baselines. Notably, both "Ours*" and our default "Ours" achieve the highest fidelity, demonstrating that our method generalizes well across models and improves quality through effective initialization and optimization.

**Ablation on texture interpolation**    In Fig. 7 (rows 4 & 5), we evaluate the effectiveness of texture controlled morphing enabled by our selective interpolation strategy. In row 4, directly increasing source token weights via naïve linear interpolation leads to texture collapse, with disorganized colors and unclear semantics. In contrast, row 5 shows that our selective interpo-

Table 3: **Ablation on texture interpolation.**

| Model | FID ↓ | STP-GPT ↑ | SEP-GPT ↑ | PPL ↓ | V-CLIP ↑ |
|---|---|---|---|---|---|
| w/o interp | 6.52 | 0.83 | 0.85 | 3.04 | 0.81 |
| Linear | 5.17 | 0.92 | 0.90 | 2.95 | 0.86 |
| **Ours** | **4.20** | **1.00** | **1.00** | **2.91** | **0.90** |

lation yields smooth morphing results while preserving distinctive source features like Wukong's appearance and attire. This demonstrates that selective control is crucial for high-quality and identity-preserving texture transitions.

We conduct quantitative ablation experiments for texture interpolation strategy as shown in Tab 3. The first row "w/o interp" refers to directly copying the source texture tokens across all steps without any interpolation. The second row "linear" refers to applying standard linear interpolation between source and target texture tokens. The third row "ours" corresponds to using our proposed selective interpolation strategy based on similarity thresholding. Our method achieves the highest visual fidelity and semantic consistency throughout the morphing trajectory, demonstrating that our method generalizes effectively across various 3D generative models. Additionally, it significantly enhances interpolation quality by carefully designing both the texture interpolation procedures.

# 5    Conclusion

We present a unified and flexible framework, WUKONG, for high-quality 3D morphing driven by minimal input—either in the form of image or text prompts. By leveraging a rectified flow-based generative model as a prior, our method enables semantically meaningful and structurally consistent shape and texture transitions. We reformulate the interpolation process using an optimal transport barycenter approach, and further enhance its stability and realism through a sequential initialization strategy. Additionally, our selective texture interpolation module offers fine-grained control over appearance, allowing users to preserve or blend semantic attributes as needed. Extensive experiments across diverse categories confirm the effectiveness of our design, with our method consistently outperforming prior state-of-the-art in both shape fidelity and texture consistency.

**Acknowledgments**    This work is supported by Hong Kong Research Grant Council - General Research Fund (Grant No. 17213825). We would like to thank Tianshuo Yan for the invaluable help during the paper preparation.

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

# Wukong's 72 Transformations: High-fidelity Textured 3D Morphing via Flow Models
# – *Appendix* –

## A  Model details

To enable high-fidelity textured 3D morphing, we build upon two pretrained flow-based transformer models introduced in Trellis (Xiang et al., 2024): the structure flow model and SLat flow model, originally designed for unconditional 3D generation. The structure flow model operates on a structured latent representation and follows a transformer-based architecture with 24 modulated transformer blocks with cross attentions. Each block contains self-attention, cross-attention, and feed-forward components, modulated via AdaLN (Guo et al., 2022) conditioning from a learned timestep embedding. Root Mean Square Normalization (RMSNorm) (Zhang and Sennrich, 2019) is applied to both the query and key representations prior to their use in the attention mechanism. The SLat flow model incorporates a hierarchical design with sparse 3D convolutional blocks and positional embeddings to encode spatial context. The transformer core comprises 24 modulated sparse transformer blocks with cross attentions, analogous in structure to the geometry model but enhanced with sparse attention and feed-forward operations. Additionally, the model includes dedicated output convolutional blocks for upsampling and decoding, ensuring fine-grained preservation and modulation of high-frequency texture details. We use the free-support Wasserstein barycenter solver from the POT library (`ot.lp.free_support_barycenter` (Lindheim, 2023)), which is based on linear programming (LP). The cost matrix is computed using the squared Euclidean distance in the CLIP (Radford et al., 2021) (for text) and DINOv2 (Oquab et al., 2023) (for image) embedding space.

## B  Ablation study

To ensure a fair, apples-to-apples comparison with 3DRM, we implemented our full morphing method on top of the GaussianAnything (Yushi et al., 2025) framework—the same 3D generator used in 3DRM (Yang et al., 2025), results are shown in Tab 4. We denote this variant as "Ours*" in the table below. Across all evaluation metrics, including FID, PPL, V-CLIP, and GPT-based perceptual scores (STP-GPT, SEP-GPT), our method consistently outperforms 3DRM, even when both share the exact same backbone. This clearly demonstrates that the improvement is not solely due to the use of a stronger generator like Trellis, but rather stems from our core morphing algorithm. Note that the GPT-based results may differ from those presented in the main paper, as they are computed using only the four methods reported here.

Table 4: **Quantitative comparison with GaussianAnything as backbone.**

| Model | FID ↓ | STP-GPT ↑ | SEP-GPT ↑ | PPL ↓ | V-CLIP ↑ |
|---|---|---|---|---|---|
| MorphFlow | 147.70 | 0.38 | 0.41 | 3.10 | 0.78 |
| 3DRM | 6.36 | 0.85 | 0.80 | 3.02 | 0.84 |
| Ours* | 5.15 | 0.93 | 0.91 | 2.94 | 0.87 |
| **Ours** | **4.01** | **1.00** | **1.00** | **2.91** | **0.90** |

Besides, we conduct quantitative evaluations with different threshold $\tau$ values and present the results below as shown in Tab 5. We observe that the performance is robust across a reasonable range of thresholds (0.2-0.8). We set a default threshold 0.3 in our evaluation.

Table 5: **Ablation study on threshold $\tau$.**

| Threshold $\tau$ | 0.2 | 0.3 | 0.4 | 0.6 | 0.8 |
|---|---|---|---|---|---|
| FID ↓ | 4.54 | 4.20 | 4.17 | 4.25 | 4.49 |
| PPL ↓ | 2.94 | 2.91 | 2.91 | 2.92 | 2.93 |
| V-CLIP ↑ | 0.88 | 0.90 | 0.91 | 0.90 | 0.87 |

To evaluate the method's generalization to real 3D data, we conducted experiments using the Headspace dataset (Dai et al., 2020), which contains high-quality 3D face scans along with corresponding rendered RGB images. In our pipeline, we used these rendered images as inputs and passed them through the DINOv2 (Oquab et al., 2023) encoder to extract texture and semantic features for morphing. The outputs were generated by our standard pipeline without any architectural changes or fine-tuning. Despite relying on pretrained components, our method shows strong generalization to real-world 3D scans. Our method outperformed both MorphFlow (Tsai et al., 2022) and 3DRM (Our own implementation) (Yang et al., 2025) on the same evaluation protocol. Quantitative results are shown below in Tab 6:

Table 6: **Quantitative results on Headspace dataset.**

| Model | FID ↓ | STP-GPT ↑ | SEP-GPT ↑ | PPL ↓ | V-CLIP ↑ |
|---|---|---|---|---|---|
| MorphFlow | 95.24 | 0.53 | 0.47 | 3.22 | 0.84 |
| 3DRM | 6.61 | 0.83 | 0.77 | 3.04 | 0.88 |
| **Ours** | **3.97** | **1.00** | **1.00** | **2.88** | **0.96** |

## C  Rectified flow models *vs.* diffusion models

**Continuity**   The first reason we choose the flow model over the diffusion model for 3D morphing is its mathematically grounded continuity with respect to the interpolation parameter $\alpha$. In flow-based generative models, the mapping from $\alpha$ to the output $F(\alpha)$ is deterministic and constructed via an invertible transformation $T(z; \alpha)$, typically defined by an ordinary differential equation (ODE). Under standard regularity conditions (e.g., Lipschitz continuity of the velocity field), the solution $T(z; \alpha)$ is guaranteed to be continuously differentiable with respect to $\alpha$ (Loud, 1987), ensuring that the morphing trajectory forms a smooth path in the output space. This deterministic nature makes it possible to precisely control intermediate shapes and textures, yielding consistent and artifact-free transitions.

In contrast, diffusion models are typically governed by stochastic differential equations (SDEs), which introduce randomness throughout the generative process. While deterministic sampling methods like DDIM (Song et al., 2021) exist and are widely used, the underlying denoising process in these models often follows a stochastic trajectory. Consequently, even with interpolated conditioning, the same value of $\alpha$ can yield different outputs across runs. This inherent variability makes it difficult to ensure continuity or precise control in the morphing sequence, particularly at intermediate points where uncertainties compound. In contrast, rectified flow models generate deterministic and unique interpolation paths, enabling forward integration with a guaranteed likelihood formulation. This property makes them more suitable for achieving smooth, stable, and controllable interpolation, which aligns with our need for consistency in the latent space.

**Convexity and optimality**   There are theoretical guarantees for flow models like the rectified flow model in maintaining the convexity of data during the generation process. This linearity ensures that any intermediate sample lies within the convex hull of the endpoints, thereby preserving the convexity of the data. In practice, the trajectory is hard to remain straight. There is analysis (Liu et al., 2023) on the straightness error on the trajectory, which states that even an imperfect trajectory is close enough to straight lines and ensures the convexity of data to some extent. Theorems in this analysis further emphasize the uniqueness and optimality of the solution of rectified flow in matching distributions under convex cost functions. For diffusion models, the backward process generates data from the prior but does not theoretically guarantee convexity preservation. These models focus on matching data distributions, not preserving geometric properties like convexity. There is no theoretical guarantee on the data convexity in the backward process. The noise term in the reverse-time SDE can easily violate the convexity of the original data. Also, under the same condition as the rectified flow model, the path of diffusion models is not assured to be optimal. There exist certain crossing flows in the matching of two distributions, leading to features that are out-of-distribution in practice.

**Inference speed**   The theoretical basis for the faster inference of flow models (such as rectified flow models) primarily stems from the geometric properties of their trajectories and the efficiency of numerical simulation. For rectified flow model used in this paper, it aims to make generation

Table 7: **Quantitative comparison with shape morphing methods.**

| Metrics | MapTree | BIM | SmoothShells | NeuroMorph | SRIF | Ours |
|---|---|---|---|---|---|---|
| Dirichlet ↓ | 17.7309 | 12.4723 | 14.0198 | 22.0461 | 6.4702 | **4.5163** |
| Cov. ↑ | 0.3967 | 0.4665 | 0.6275 | 0.1099 | 0.6418 | **0.8510** |

trajectories as straight as possible. For ideal straight-line flows, the trajectory between any two points $Z_0 \sim \pi_0$ and $Z_1 \sim \pi_1$ is given by the linear interpolation $Z_t = tZ_1 + (1-t)Z_0$. In this case, the drift field of the ODE is a constant $v(Z_t, t) = Z_1 - Z_0$, which can be solved exactly with a single Euler step: $Z_1 = Z_0 + v(Z_0, 0) \cdot 1$. This eliminates the need for time discretization errors. Even in non-ideal cases, the optimized trajectories are close to straight lines, significantly reducing the number of required steps (in this paper, we take 20 steps). Diffusion models use nonlinear, stochastic trajectories requiring many steps—typically 2,000 without sampling techniques or around 200 with them—to achieve good results. Also, the reverse SDE process requires noise sampling, leading to additional computation cost.

## D    Comparison with other methods

### D.1    Comparison with other textured 3D method

We compare our method with existing textured 3D morphing approaches, including MorphFlow, 3DRM, and our own. While the main paper presents qualitative comparisons with 3DRM, here we additionally provide a side-by-side qualitative comparison with MorphFlow. As shown in the Fig. 9, rows 1 and 3 display results from MorphFlow, and rows 2 and 4 show our corresponding outputs. Our method produces noticeably clearer and more accurate shapes and textures, with smoother and more coherent morphing transitions. These visual improvements are consistent with the quantitative results reported in the main paper, further corroborating the effectiveness of our approach.

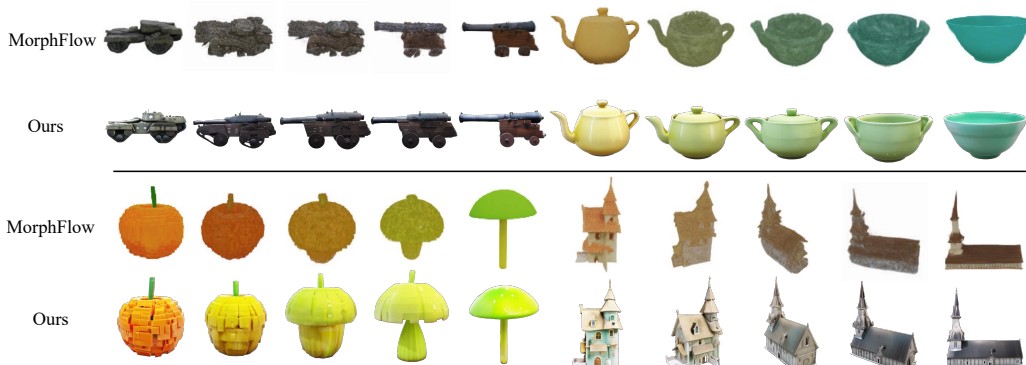

Figure 9: **Qualitative comparison with MorphFlow.**

### D.2    Comparison with shape morphing methods

Although previous 3D shape morphing methods do not consider texture transformation, we provide a comparison focused solely on shape deformation. As shown in Tab 7, we compare our method with several state-of-the-art shape morphing approaches, including MapTree (Ren et al., 2020), BIM (Kim et al., 2011), SmoothShells (Eisenberger et al., 2020), NeuroMorph (Eisenberger et al., 2021), and SRIF (Sun et al., 2024). Following the evaluation protocol from (Sun et al., 2024), we use the SHREC07 (Temerinac et al., 2007) dataset and report performance using Dirichlet energy (Ezuz et al., 2019) and Coverage (Huang and Ovsjanikov, 2017) metrics. Our method achieves superior results across both metrics, demonstrating more efficient and accurate shape interpolation.

For qualitative comparison, we show results against SRIF in Fig. 10, where rows 1, 3, and 5 show SRIF's outputs and rows 2, 4, and 6 show ours. Our morphing process is smoother and preserves finer details in intermediate shapes—for example, the gecko's toes in row 2 and the head structure in row 4. Additional comparison with NeuralMorph is presented in Fig. 11, where our results are again significantly more detailed and coherent.

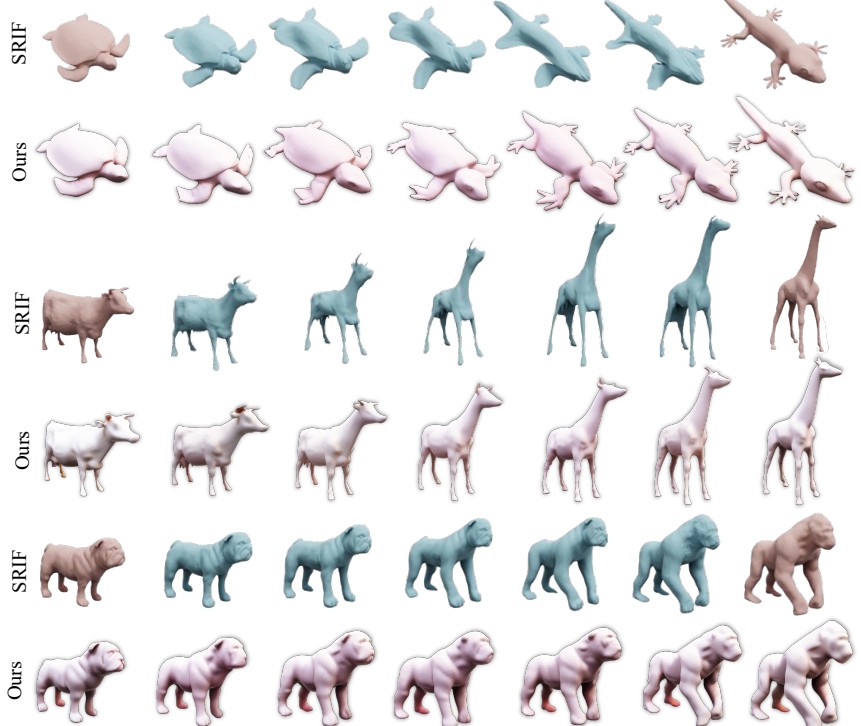

Figure 10: **Qualitative comparison with SRIF.**

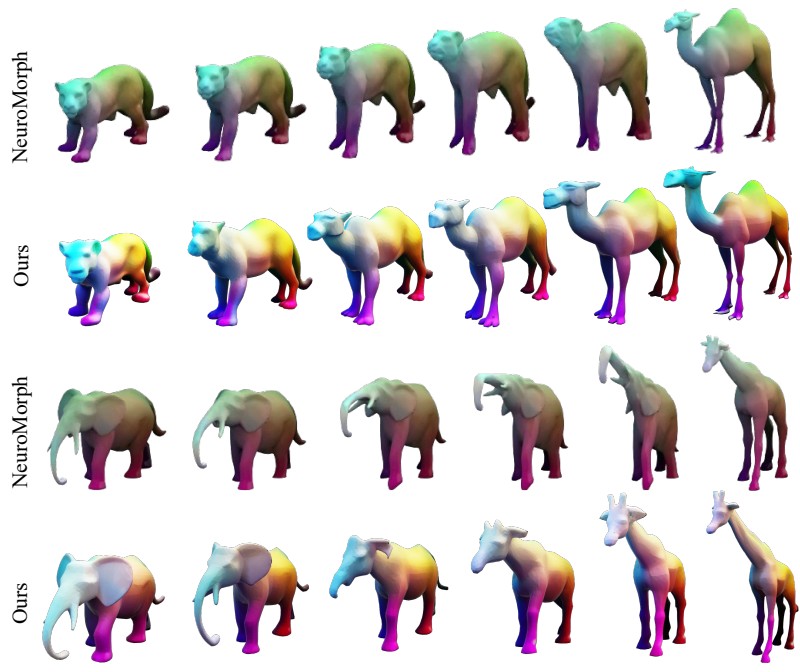

Figure 11: **Qualitative comparison with NeuralMorph.**

To ensure that intermediate shapes remain faithful to both source and target, we introduce a shape-aware initialization. Specifically, we render both front and back views of the source and target objects and use these images as inputs to the flow model to extract an initial condition feature. This feature is then refined by minimizing the geometric difference between generated shapes and the original meshes, leading to accurate and consistent 3D representations throughout the morphing sequence.

## E  More results

Fig. 12 and Fig. 13 illustrate the textured 3D morphing process generated from "Wukong" to a variety of objects. The texture can be flexibly inherited from either the source or the target image, depending on the user's preference. Fig. 14 and Fig. 15 further demonstrate morphing between additional object pairs, showcasing the versatility of our method. Notably, our method is capable of performing textured 3D morphing not only between geometrically complex objects, but also across different semantic categories, highlighting its superior robustness and generalizability.

## F  Broader impact

Our work introduces WUKONG , a training-free framework for high-quality textured 3D morphing, which significantly lowers the barrier to creating detailed and semantically consistent 3D transformations from simple prompts. This greatly reduces the efforts on 3D content creation for artists, designers, and educators, enabling broader access to advanced generative tools without requiring technical expertise in 3D modeling or animation. The ability to produce controllable and high-fidelity morphing sequences could benefit applications in virtual reality, digital storytelling, education, and creative industries. We hope our work inspires further research into controllable and efficient 3D generation techniques, and that it serves as a foundation for inclusive and creative applications of generative 3D content.

## G  Limitation

While our method achieves state-of-the-art performance in textured 3D morphing, several limitations remain. First, like existing morphing methods, our approach still encounters difficulties in cases involving extreme topological changes, such as splitting or merging parts. These scenarios remain a general challenge in the field and are not yet fully addressed by existing methods. Second, since our method operates without explicit 3D supervision or correspondence annotations, its results may be sensitive to ambiguities in the input prompts or inconsistencies in multi-view generation, especially when the input lacks structural clarity.

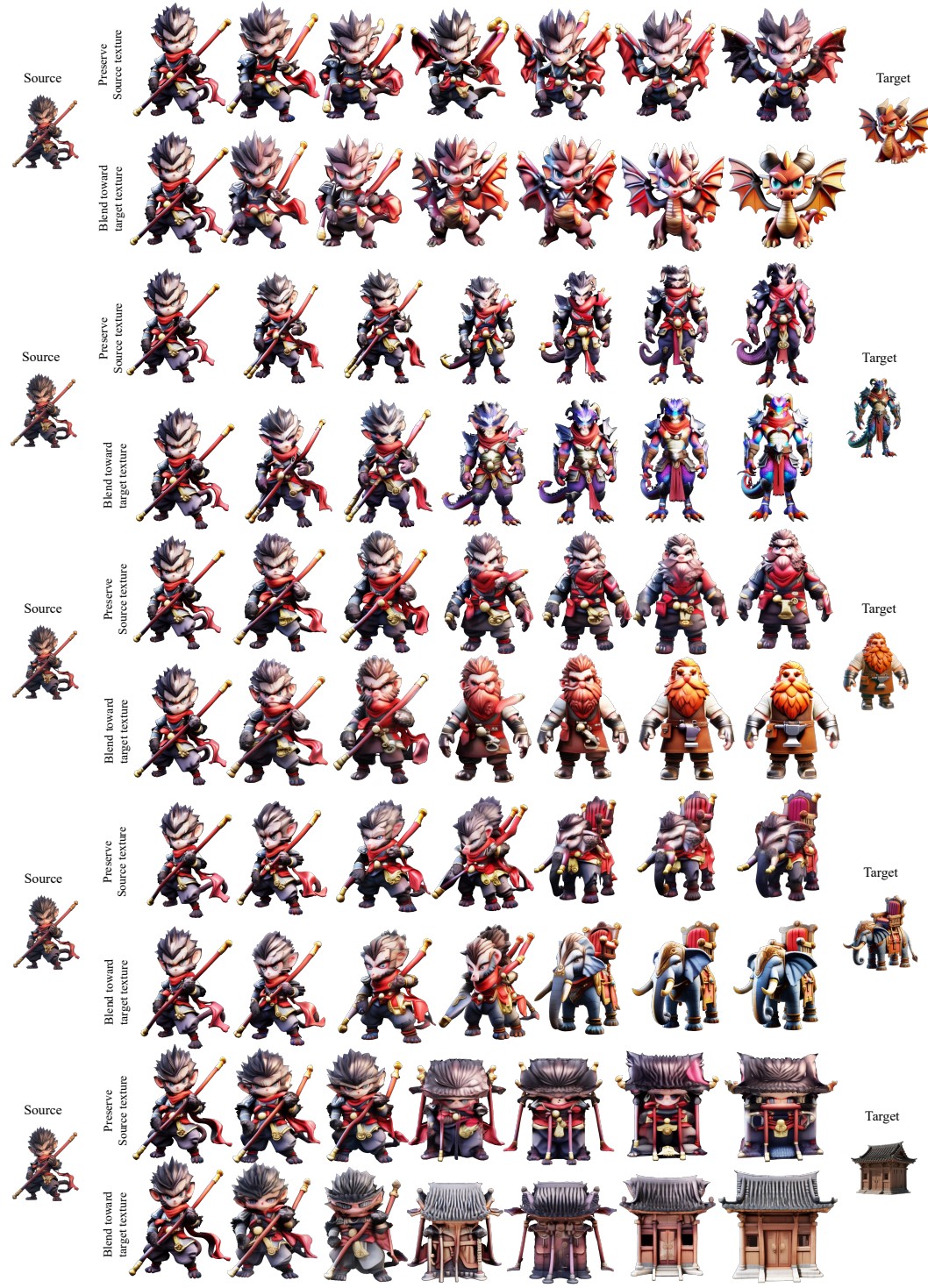

Figure 12: **Textured 3D morphing of Wukong (The Monkey King).**

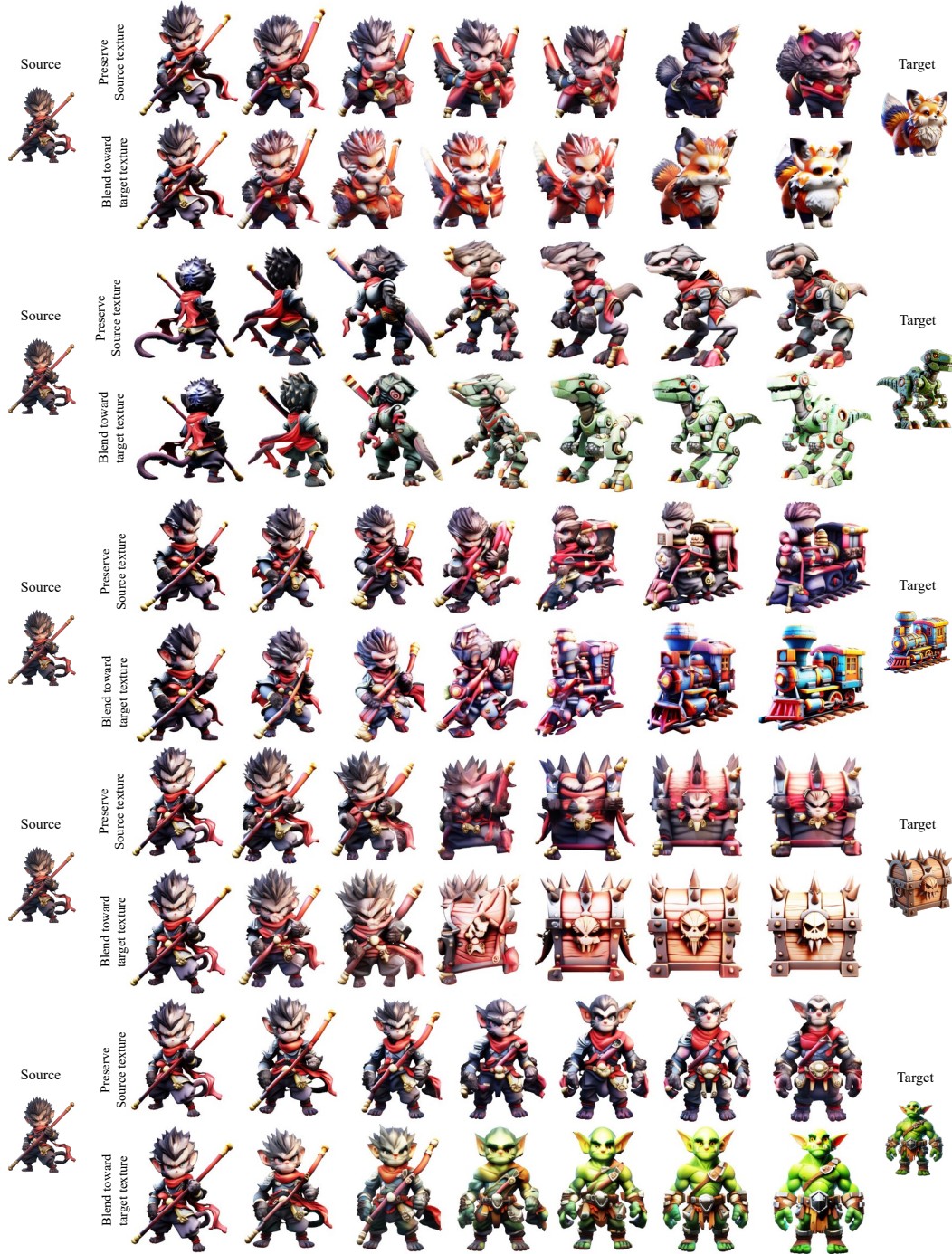

Figure 13: **Textured 3D morphing of Wukong (The Monkey King).**

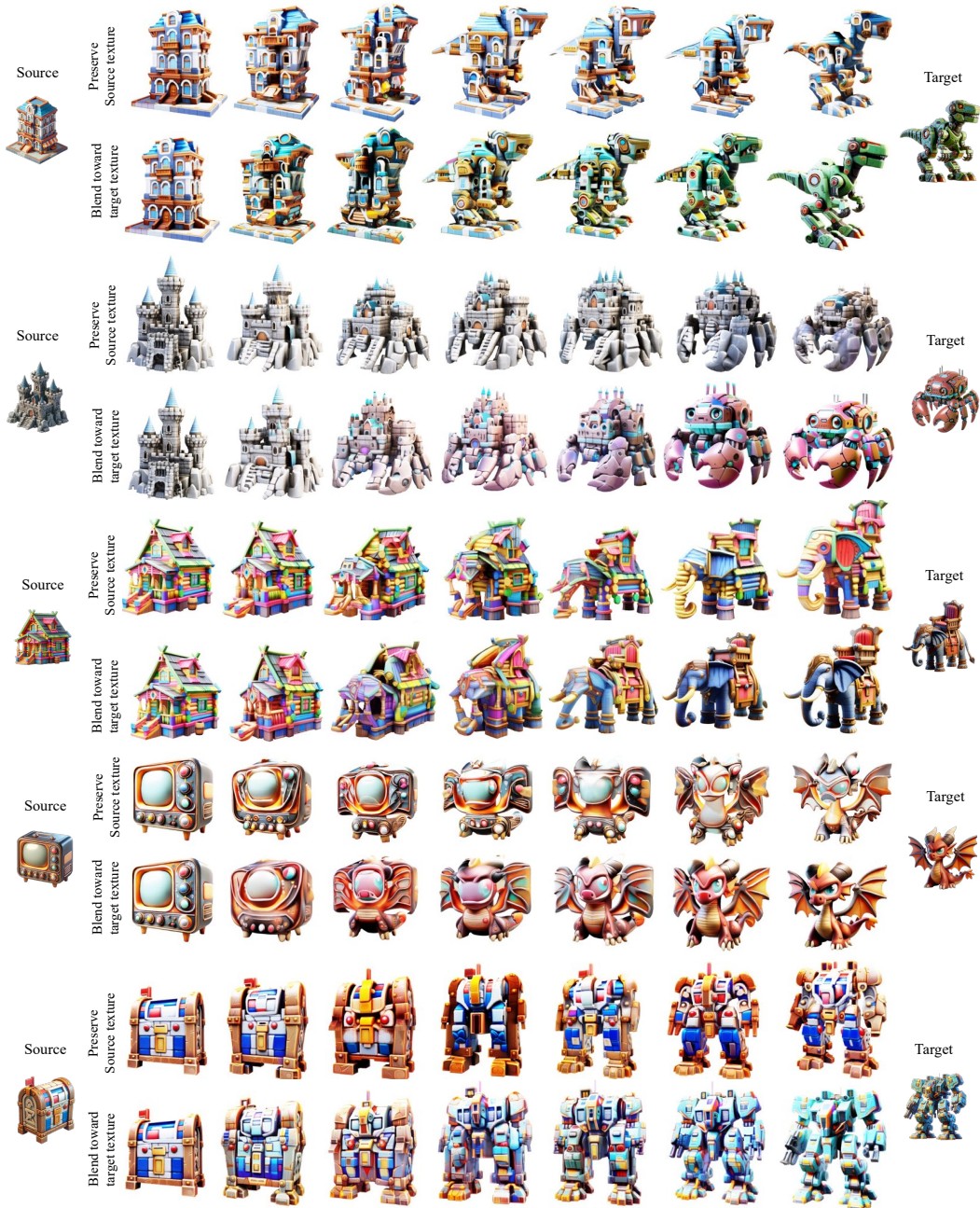

Figure 14: **Textured 3D morphing of different objects.**

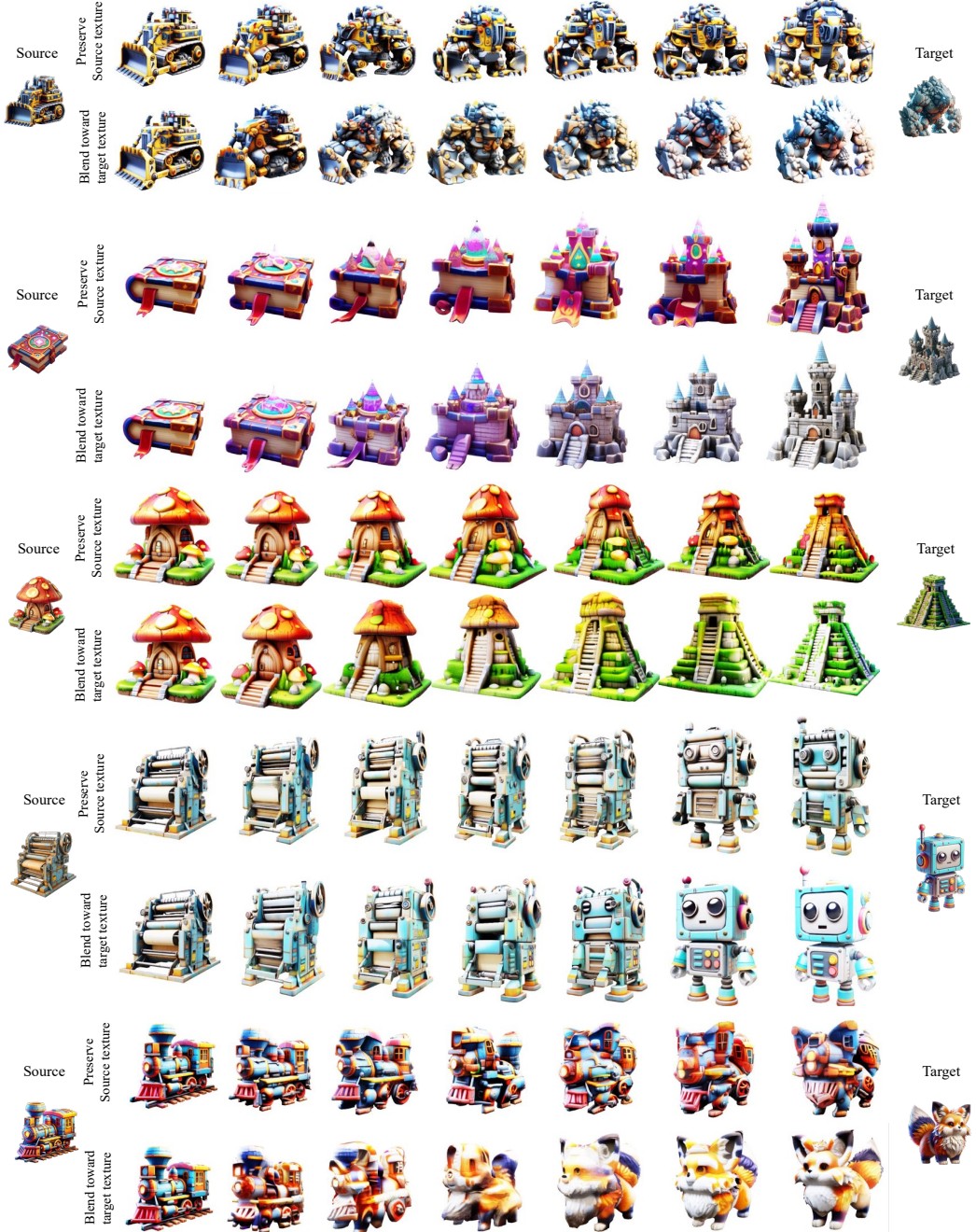

Figure 15: **Textured 3D morphing of different objects.**

