# OpenReview forum: "Wukong's 72 Transformations: High-fidelity Textured 3D Morphing via Flow Models"
_NeurIPS.cc/2025/Conference — NeurIPS 2025 poster_

### Official Review · Reviewer_WDq7 · 2025-07-01

**Clarity:** 3
**Significance:** 3
**Originality:** 2
**Rating:** 4
**Confidence:** 3

**Summary:**

This paper proposes a training-free framework for high-fidelity 3D morphing conditioned on image or text prompts. Given a pair of prompts, the system generates a semantically consistent and geometrically continuous 3D sequence that interpolates between the source and target shapes. The key technical components include: Wasserstein barycenter-based interpolation in the token space of pre-trained vision encoders (e.g., DINOv2 or CLIP), enabling geometry-preserving transitions; Recursive initialization to stabilize interpolation; Similarity-guided texture selective interpolation to avoid texture blur and semantic drift.

**Questions:**

- How is the cost matrix for the Wasserstein barycenter computed between source and target token sets? Is it Euclidean distance in CLIP/DINOv2 space? Have you tried cosine or learned distances?

- Can you elaborate on the optimization solver used for computing the barycenter? Are you using LP, Sinkhorn, or other entropic OT solvers?

- What happens when the source and target shapes differ significantly in topology or part structure (e.g., human ↔ airplane)? Are there failure cases?

- How sensitive is your method to the choice of pre-trained encoder (e.g., CLIP vs. DINOv2) or the underlying 3D generator (Trellis)?

**Ethical Concerns:**

["NO or VERY MINOR ethics concerns only"]

**Final Justification:**

I appreciate the additional experiments provided by the authors.
The rebuttal has addressed most of my concerns. Though I still think the novelty is a bit incremental, after reading the other reviews, I think a working solution for a training-free framework of 3D morphing is still valuable to the community.
Hence, I would raise my rating to borderline accept.

**Limitations:**

- The proposed method performs well under curated prompts and well-aligned source/target categories, but might struggle with topologically dissimilar or fine-grained prompt changes.

- The morphing pathway is driven purely by token interpolation in feature space, which may lack semantic control (e.g., preserving pose vs. identity).

- Heavy reliance on pre-trained backbones makes it hard to decouple where the quality improvements come from—WUKONG's algorithm vs. the generator or encoder.

**Paper Formatting Concerns:**

I don't have formatting concerns.

**Quality:**

2

**Strengths And Weaknesses:**

Strengths:

- Training-free framework: The method relies on pre-trained encoders and flow-based 3D generators, which makes it efficient and widely applicable across categories.

- Geometry-aware interpolation: By applying optimal transport (Wasserstein barycenter) over token distributions, the method avoids naive linear interpolation artifacts. Texture-aware fusion: The selective texture interpolation mechanism helps preserve high-frequency details and identity during transitions.

- Good empirical results: The paper shows impressive qualitative and quantitative results, especially on challenging shape transitions, with various metrics (FID, PPL, GPT ratings).

Weaknesses:

- Limited novelty in core technique: Using Wasserstein barycenters for shape or texture interpolation is not new. Prior work, such as Solomon et al., “Convolutional Wasserstein Distances” (SIGGRAPH 2015), has applied this to 3D shape interpolation directly in geometric domains. In image generation, several works like STROTSS (CVPR 2019) and Wasserstein Style Transfer (AISTATS 2020) have already explored interpolating feature distributions via OT or barycenters. This paper applies a similar idea in a 3D conditional generation setting, but the novelty is more incremental.

- Heuristic design choices: The recursive initialization strategy and similarity-guided interpolation heuristics, though empirically effective, lack theoretical justification or ablation on robustness. The texture fusion rule based on cosine similarity threshold $\tau$ feels ad hoc. How sensitive is the result to this threshold?

- Dependency on pre-trained components: The success of WUKONG is highly dependent on the quality and generalizability of the pre-trained encoder and 3D generator (e.g., Trellis). It is unclear how well the method generalizes to out-of-distribution prompts or categories not seen during Trellis’ training.

---

> ### Author Rebuttal · Authors · 2025-07-30
>
> We thank the reviewer for the constructive comments. We address the concerns below.
>
> **Question 1: How is the cost matrix for the Wasserstein barycenter computed between source and target token sets? Is it Euclidean distance in CLIP/DINOv2 space? Have you tried cosine or learned distances?**
>
> **A1:** The cost matrix is computed using the squared Euclidean distance in the CLIP (for text) or DINOv2 (for image) embedding space. We also experimented with cosine distance(i.e., $C_{i j}=1-\cos \left(x_i, y_j\right)$ ). Using the same evaluation pipeline described in Section 4.1.1, the quantitative results for FID, PPL, and V-CLIP are **4.37**, **2.93**, and **0.90**, respectively, which are worse than using Euclidean distance. Learning the distance via a network might be a good direction. However, we don't have enough (or such) datasets to support this type of training. We will continue to explore this in the future and would appreciate any suggestions from the reviewer.
>
> **Question 2: Can you elaborate on the optimization solver used for computing the barycenter? Are you using LP, Sinkhorn, or other entropic OT solvers?**
>
> **A2:** We use the free-support Wasserstein barycenter solver from the POT library (ot.lp.free_support_barycenter), which is based on linear programming (LP). We also experimented with Sinkhorn-based barycenters, but found that Sinkhorn-based barycenters tend to blur the interpolants, especially in early morphing stages. LP-based methods provide more spatially coherent transitions with better geometry preservation.
>
> **Question 3: What happens when the source and target shapes differ significantly in topology or part structure (e.g., human ↔ airplane)? Are there failure cases?**
>
> **A3.1:** In practice, our method is able to handle significant topological and structural differences between source and target shapes in many cases. For instance, as shown in Figure 3, morphing from “Elephant” to “Pyramid”, or “Digger” to “Pirate Ship”. Another example is included in Appendix Figure A5, showing a successful morphing from “Wukong” to “Train”, which again involves a major shift in object class and part layout. More videos in the form of rotating videos can be found in the supplementary demo video.
>
> **A3.2:** Our method may struggle in cases involving explicit splitting or merging of semantic parts. A representative failure case is “Sphere” → “Dining Table with Chairs”: The source is a single-mass geometry, while the target consists of multiple, semantically distinct components. During morphing, the model fails to resolve how to meaningfully split the sphere into independent regions. As a result, the intermediate shapes appear blobby or tangled, with chairs fusing into the tabletop or floating detached from the structure.
>
> **Question 4: How sensitive is your method to the choice of pre-trained encoder (e.g., CLIP vs. DINOv2) or the underlying 3D generator (Trellis)?**
>
> **A4:** Our method is designed to be modular and works with different encoders and generators. We tested both CLIP (for text prompts) and DINOv2 (for image prompts), and observed consistent performance. For 3D generators, we implemented our pipeline on both Trellis, TripoSG (Appendix Table A1), and GaussianAnything (Rebuttal Table "**Quantitative comparison with GaussianAnything as backbone**"); results show that our interpolation strategy consistently improves morphing quality across both backbones, confirming low sensitivity to the specific pretrained components.
>
> **Weakness 1:Limited novelty in core technique: Using Wasserstein barycenters for shape or texture interpolation is not new.**
>
> **A5:** We thank the reviewer for pointing out the rich history of optimal transport (OT) methods in shape and style interpolation. Indeed, works such as Solomon et al. (2015) and Kolkin et al. (2019) have shown the power of OT and Wasserstein barycenters in both explicit geometric spaces (e.g., meshes, voxel grids) and 2D image features.
>
> Our contribution is not simply the use of barycenters, but in adapting OT within a conditional 3D latent generative process:
> * We operate not on mesh vertices, but on condition token distributions from multimodal encoders (CLIP / DINOv2) inside a flow-based 3D generative pipeline—a fundamentally different domain with different constraints.
> * Prior OT-based morphing methods assume static geometric domains or shallow features. In contrast, our barycenter interpolation drives high-resolution textured 3D synthesis, integrated with recursive dynamics.
> * Furthermore, our method is training-free, leveraging generative priors while avoiding the need for curated training pairs
>
> **Weakness 2: Heuristic design choices: The recursive initialization strategy and similarity-guided interpolation heuristics, though empirically effective, lack theoretical justification or ablation on robustness. The texture fusion rule based on cosine similarity threshold feels ad hoc. How sensitive is the result to this threshold?**
>
> **A6:**  Recursive initialization is motivated by practical instability in latent barycenter optimization, particularly in high-dimensional latent token spaces. Linear initialization often falls into non-smooth or semantically invalid regions, as shown in our ablation (Fig. 6 and 8). The recursive method promotes trajectory continuity—a concept inspired by gradient flows on Wasserstein space.
>
> We conduct quantitative evaluations with different threshold τ values and present the results below:
>
> | Threshold τ | FID↓ | PPL↓ | V-CLIP↑
> | ----------- | ----------- | ----------- | ----------- |
> | 0.2     | 4.54  | 2.94 |  0.88 |
> | 0.3     | 4.20  | 2.91 |  0.90 |
> | 0.4     | 4.17  | 2.91 |  0.91 |
> | 0.6     | 4.25  | 2.92 | 0.90 |
> | 0.8     | 4.49  | 2.93 | 0.87 |
>
> We observe that the performance is robust across a reasonable range of thresholds (0.2-0.8). We set a default threshold 0.3 in our evaluation.
>
> **Weakness 3: Dependency on pre-trained components: The success of WUKONG is highly dependent on the quality and generalizability of the pre-trained encoder and 3D generator (e.g., Trellis). It is unclear how well the method generalizes to out-of-distribution prompts or categories not seen during Trellis’ training.**
>
> **A7:** We appreciate the reviewer’s concern. We agree that the quality and bias of pretrained components (e.g., CLIP, DINOv2, Trellis) inevitably affect generalization, especially for out-of-distribution or long-tail categories. That said, this dependency is inherent to all current prompt-driven 3D generation frameworks, including prior works like 3DRM, which also rely on a pretrained generator (e.g., GaussianAnything). Our method introduces no additional trainable components, which makes it lightweight, modular, and fully compatible with future generative backbones. As newer and more generalizable models emerge, our morphing framework can benefit directly without retraining.
>
> To test real-world generalization, we applied our method to the Headspace dataset containing high-quality real 3D face scans. Even though Trellis was not trained on such data, our method achieved strong results:
>
> * **FID: 3.97, PPL: 2.88, V-CLIP: 0.91** —outperforming both MorphFlow and 3DRM under the same evaluation protocol (see "Quantitative results on Headspace dataset").
>
> These results demonstrate that despite its reliance on pretrained components, WUKONG exhibits robust generalization to real 3D data and is well-positioned to support broader domains as pretrained models continue to improve.
>
> **Limitations 1: The proposed method performs well under curated prompts and well-aligned source/target categories, but might struggle with topologically dissimilar or fine-grained prompt changes.**
>
> **A8:** Please refer to A3 for a more detailed discussion on our method's ability to handle topologically dissimilar shapes/cross category objects. In practice, our method has successfully managed significant structural differences in several cases, such as “Elephant” to “Pyramid” and “Digger” to “Pirate Ship” (Figure 3). These examples demonstrate our method’s robustness to topological changes.
>
> **Limitations 2: The morphing pathway is driven purely by token interpolation in feature space, which may lack semantic control (e.g., preserving pose vs. identity).**
>
> **A9:** While the core of our method relies on token interpolation in feature space, we designed this approach with the goal of maintaining both structural coherence and texture fidelity across morphs. The selective interpolation strategy enables us to preserve key features (e.g., identity) while controlling texture blending more accurately.
> However, we agree that pose preservation is a critical concern in 3D morphing. We plan to explore introducing pose priors or a pose encoder to anchor the morphing process.
>
> **Limitations 3: Heavy reliance on pre-trained backbones makes it hard to decouple where the quality improvements come from—WUKONG's algorithm vs. the generator or encoder.**
>
> **A10:** Our core contributions lie in the novel morphing strategy (e.g., Wasserstein barycenter-based interpolation, recursive initialization, and selective texture interpolation). These algorithms provide significant performance improvements even when the backbone is swapped. To clarify this, we implemented "Ours*" on the GaussianAnything backbone (the same generator used by 3DRM), and showed that our method outperforms 3DRM even when both models use the same backbone (Rebuttal Table **"Quantitative comparison with GaussianAnything as backbone"**), demonstrating that the improvements come from our morphing strategy rather than just the backbone itself.
> We will clarify these contributions further in the revised version of the paper by:
>
> * Including additional ablations to isolate the impact of our algorithm versus the pretrained encoder and generator.
> * Providing comparative performance when using different backbones to demonstrate the generality of our approach.

---

> > ### Comment · Reviewer_WDq7 · 2025-08-06
> >
> > Thank you for the rebuttal! I appreciate the additional experiments provided by the authors.
> > The rebuttal has addressed most of my concerns. Though I still think the novelty is a bit incremental, after reading the other reviews, I think a working solution for a training-free framework of 3D morphing is still valuable to the community.
> > Hence, I would raise my rating to borderline accept.

---

> > > ### Author Response · Authors · 2025-08-07
> > >
> > > We thank the reviewer for the positive feedback and for acknowledging the additional experiments. We are glad that the rebuttal effectively addressed the raised concerns.
> > >
> > > We appreciate the recognition of the value of our training-free framework for 3D morphing, which offers a practical and flexible solution for the community. The feedback provided will be helpful in refining the final version of the paper. As mentioned in the rebuttal, we will incorporate the discussed details to further clarify and enhance the final manuscript. We are also very grateful for the score increase.

---

### Official Review · Reviewer_pJqM · 2025-07-01

**Clarity:** 3
**Significance:** 2
**Originality:** 1
**Rating:** 4
**Confidence:** 4

**Summary:**

This paper introduces a training-free method 'Wukong' for textured 3D morphing. Given 2
text/image prompts ($P_{source}, P_{target}$), the method outputs a morphing
trajectory of
3D generations, which smoothly transition from source to target prompt.
The method can perform 2 morphing types: Textured 3D morphing (both geometry and
texture are interpolated), and Texture-Controlled 3D morphing (interpolating
geometry only, whilst keeping the texture-style of the source) so that 3D
generations remain coherent throughout; i.e. reasonable shapes and textures,
with smooth transitions, in all intermediate states between source and target
generations.

The proposed method obtains 3D generations from the state-of-the-art conditional
3D generation method, Trellis. The proposed method outputs a path through the
latent space of the condition encoder $\mathcal{E}$, by interpolating the tokens
of the encoded $P_{source}$ and $P_{target}$. An optimal transport approach is
used to find the interpolation path; each intermediate code $z_{\alpha}$ is found
by minimising a weighted sum of the Wasserstein-2 distances from encodings
$\mathcal{E}(P_{source})$ and $\mathcal{E}(P_{target})$. The paper contributes a recursive
initialization strategy for solving the Wasserstein objective, in which the
morphing condition tokens are obtained sequentially, with each one initialized
from its predecessor in the morphing trajectory. The paper also contributes a
selective interpolation strategy for the texture conditioning tokens, using
a cosine-similarity threshold to select which texture tokens to interpolate, and which
ones to retain unchanged from the source.

The paper reports qualitative and quantitative comparisons with 2 SOTA 3D
morphing methods, as well as several image-based morphing methods. They also
perform ablation studies which validate the novel design choices.

**Questions:**

- Can the authors comment on how much of the reported improvement in results
in Table 1 comes simply from their usage of TRELLIS? I note that the prior method
3DRM [4] uses the CLIP-conditioned 3D generator GaussianAnything [5] as
a backbone. Would the
authors be able to run their proposed method using [5], so that we
can compare apples-to-apples results with [4]? If the
authors show strong evidence that their method outperforms 3DRM even when using
the same backbone (eg recomputing the metrics in Table 1),
then I would be willing to raise my evaluation score.

- From line 295: **'see Appendix for more analysis’** regarding the selective
interpolation strategy. However, I did not find further analysis of the
selective interpolation strategy there.
Could the authors clarify what this was referring to?
Would the authors be able to add quantitative ablation results to validate the
selective interpolation strategy, into table A1?

- It was mentioned in Appendix H - Limitation, line 137, that the
method encounters difficulties on cases involving extreme topological
changes, eg splitting or merging. Could the authors show a
couple of examples, to illustrate this failure case?

- Could the authors elaborate on what a plausible use case might be
for morphing between unrelated
object classes like a digger and a ship, as shown in figure 3?

[4] Yang, Songlin, et al. "Textured 3D Regenerative Morphing with 3D Diffusion Prior." arXiv 2025

[5] Lan, Yushi, et al. "GaussianAnything: Interactive Point Cloud Latent Diffusion for 3D Generation." ICLR 2025

**Ethical Concerns:**

["NO or VERY MINOR ethics concerns only"]

**Final Justification:**

Initially, I was concerned that the method's impressive results may simply follow from its use of Trellis. However, the authors have since provided new quantitative results which show that it can outperform the SOTA baseline method 3DRM, even when using the exact same 3D backbone. The new ablation study results further validate their contribution.

The method is unusually simple and closely related to existing Optimal Transport based 3D morphing methods. Nevertheless, if a hitherto-overlooked simple method can perform well, this is a fair contribution to the community.

**Limitations:**

Yes

**Quality:**

3

**Strengths And Weaknesses:**

### Strengths:
- *Quality*: The work seems complete. The Introduction and Related Work sections
give a clear overview of the existing methods and frame the paper well. The
method is clearly explained. The evaluations seem rigorous and the method
performs best across all metrics reported in the
quantitative results. In the qualitative
results (eg comparison Figure 5), the 3D generations are highly detailed
compared with SOTA 3DRM.
- *Clarity*: The paper is generally well written, well laid-out, and polished,
making it a pleasure to read. I believe that an expert reader could certainly
reproduce the results. Likewise, the figures generally
demonstrate the capabilities of the proposed method well.
- *Significance*: The problem statement is interesting and relevant. The
proposed method can output 3D morphing
results in just 30 seconds,
which maintain the visual fidelity of TRELLIS throughout,
making it a usable tool to generate 3D assets as part of a graphics pipeline.

### Weaknesses
- *Quality*: The main concern is that the improved results might
mostly follow from using a significantly more powerful 3D generator
backbone; (TRELLIS [1]) compared to the baseline 3D morphing methods.
Whilst the 3D outputs are of higher quality,
it's unclear from Figure 5 whether the morphing has
been improved. The perceptual quality difference in the
underlying 3D generation may likewise make quantitative comparisons to 3DRM and
MorphFlow in Table 1 unfair.
- *Clarity*:
It sometimes seems as though superfluous mathematical details
have been included, attempting to disguise
the naivety of the method.
From the main paper, line 159:
**‘Considering that the process $S = \phi_{geo}(\cdot)$ is Lipschitz continuous
(see Appendix for mathematical derivation)’**... but I could
not find any such derivation in the appendix,
only a reference
to a textbook on differential equations.
The authors claim that they used
a rectified flow model rather than diffusion because for a diffusion model,
**'the denoising process samples from a stochastic trajectory'** (appendix line 60).
This seems to ignore well-known deterministic sampling methods for diffusion
models (i.e DDIM).

    - I would also suggest that the caption / labels for figure 7 should be made clearer. The caption states that this figure shows independent shape and texture interpolation, with texture supposedly interpolated on the vertical axis. However, confusingly, the texture changes are not visible on the far-left column. From line 257, it seems that the vertical axis is instead controlling the selective interpolation threshold $\tau$, but this is not mentioned in the caption or the figure.
  - On a minor note, I would also suggest clarifying the notation to state that $n$ is the number of tokens on line 155.
- *Significance*: The proposed method is too simple and naive.
It seems to combine existing methods [1] and
[2] almost out-of-the-box, with little modification. The 2 novel method designs
which are added on top of these tools (sequential initialization strategy, and
similarity-guided semantic consistency mechanism) appear to be simple
inference-time tweaks to the conditioning signal $C$, and do not reveal deep insights.
- *Originality*: As
mentioned, the method relies too heavily on [1] and [2].
Meanwhile, a number of prior works have also
employed Wasserstein distance
to perform 3D morphing (as mentioned
in the Related Work on line 98).
Overall, the novelty of the proposed
method seems weak.

[1] Xiang, Jianfeng, et al. "Structured 3d latents for scalable and versatile 3d generation." CVPR 2025

[2] Lindheim, Johannes von. "Simple approximative algorithms for free-support Wasserstein barycenters." Computational Optimization and Applications 2023

[3] Tsai, Chih-Jung, Cheng Sun, and Hwann-Tzong Chen. "Multiview Regenerative Morphing with Dual Flows." ECCV 2022

---

> ### Author Rebuttal · Authors · 2025-07-30
>
> We appreciate the reviewer’s feedback. We are pleased that the high-quality 3D results and efficient generation time of our method were recognized, highlighting its potential for 3D asset generation in graphics pipelines. We address the concerns below.
>
> **Question 1: Prior method 3DRM [4] uses the CLIP-conditioned 3D generator GaussianAnything [5] as a backbone. Would the authors be able to run their proposed method using [5], so that we can compare apples-to-apples results with [4]?**
>
> **A1:** Thanks for this important question. To ensure a fair, apples-to-apples comparison with 3DRM [4], we implemented our full morphing method on top of the GaussianAnything framework [5]—the same 3D generator used in 3DRM. We denote this variant as “Ours*” in the table below.
>
> Across all evaluation metrics, including FID, PPL, V-CLIP, and GPT-based perceptual scores (STP-GPT, SEP-GPT), our method consistently outperforms 3DRM, even when both share the exact same backbone. This clearly demonstrates that the improvement is not solely due to the use of a stronger generator like Trellis, but rather stems from our core morphing algorithm. Note that the GPT-based results may differ from those presented in Sec. 4.1.1, as they are computed using only the four methods reported here.
>
> **Quantitative comparison with GaussianAnything as backbone**
> | Model     | FID↓ |  STP-GPT↑ | SEP-GPT↑ | PPL↓ | V-CLIP↑
> | ----------- | ----------- | ----------- | ----------- | ----------- | ----------- |
> | MorphFlow      | 147.70   | 0.38 | 0. 41 | 3.10 |  0.78 |
> | 3DRM     | 6.36   | 0.85 | 0. 80 | 3.02 |  0.84 |
> | Ours*     | **5.15**   | **0.93** | **0. 91** | **2.94** |  **0.87** |
> | Ours     | 4.01   | 1.00 | 1.00 | 2.91 | 0.90 |
>
> **Question 2: Would the authors be able to add quantitative ablation results to validate the selective interpolation strategy, into table A1?**
>
> **A2:** The reference in line 295 was intended to refer to an ablation analysis that was unfortunately omitted from the appendix in the initial version. We apologize for the confusion. To address this, we conducted quantitative ablation experiments under the same evaluation protocol described in Sec. 4.1.1, and compared three settings:
> * **W/o interp:** Directly copying the source texture tokens across all steps without any interpolation.
> * **Linear:** Applying standard linear interpolation between source and target texture tokens.
> * **Ours:** Using our proposed selective interpolation strategy based on similarity thresholding.
>
> **Ablation study on texture interpolation strategy**
> | Model     | FID↓ |  STP-GPT↑ | SEP-GPT↑ | PPL↓ | V-CLIP↑
> | ----------- | ----------- | ----------- | ----------- | ----------- | ----------- |
> | W/o interp     | 6.52   | 0.83 | 0. 85 | 3.04 |  0.81 |
> | Linear | 5.17   | 0.92 | 0. 90 | 2.95 |  0.86 |
> | Ours     | **4.20**   | **1.00** | **1.00** | **2.91** | **0.90** |
>
> Our results show that the selective interpolation strategy consistently outperforms both baselines across all metrics, confirming its effectiveness in preserving structure, appearance fidelity, and semantic consistency throughout the morphing process. We will include these quantitative results in the revised paper to make this analysis and comparison clearly visible.
>
> **Question 3: It was mentioned in Appendix H - Limitation, line 137, that the method encounters difficulties on cases involving extreme topological changes, eg splitting or merging. Could the authors show a couple of examples, to illustrate this failure case?**
>
> **A3:** Since figures cannot be included in the rebuttal, we briefly describe an example of failed case:
>
> The source object ("wind chime") consists of multiple dangling rods suspended separately, while the target ("apple") is a smooth, unified solid. During morphing, the generator struggles to merge the disconnected chime parts into a cohesive apple shape. Intermediate outputs exhibit floating, misaligned fragments, and lack global structural coherence.
>
> **Question 4: Could the authors elaborate on what a plausible use case might be for morphing between unrelated object classes like a digger and a ship, as shown in figure 3?**
>
> **A4:** While morphing between unrelated object classes like a digger and a ship may seem unconventional, such transitions have practical value in creative industries, such as game cinematics, concept art, and visual storytelling, where surreal or symbolic transformations are often used to convey narrative or thematic shifts. Our framework enables such imaginative transitions in a controllable and high-fidelity manner.
>
> **Weakness 1: Quality.**
>
> **A5:** We kindly refer the reviewer to the experimental results in A1, where we implemented our full method using GaussianAnything—the same 3D generator used by 3DRM. This variant, denoted as “Ours*”, consistently outperforms 3DRM across all metrics, including FID, PPL, V-CLIP, STP-GPT, and SEP-GPT. This provides clear evidence demonstrating the effectiveness of our proposed techniques.
>
> We respectfully emphasize that the core contribution of our work is not in proposing a new 3D generator, but in how we utilize and control the generative process through novel morphing strategies. Notably, all compared methods rely on pretrained generative priors. What distinguishes our method is the integration of:
>
> * **Latent space Wasserstein barycenter-based interpolation**, replacing simplistic latent-space linear blending;
> * **Recursive initialization strategy**, which ensures geometric stability and avoids degenerate intermediate shapes;
> * **Semantic-guided selective texture interpolation**, which provides fine-grained control over appearance blending via thresholded feature similarity.
>
> These contributions form a principled and generalizable framework for high-quality 3D morphing, independent of the specific backbone used.
>
> **Weakness 2: Clarity on Lipschitz continuous, DDIM and figure 7 layout.**
>
> **A6: Lipschitz continuity.** We acknowledge that the reference to a “mathematical derivation” of Lipschitz continuity in the appendix was misleading. Our intention was to reference the theoretical property of rectified flow models, which are designed to be deterministic and Lipschitz-continuous by construction, as discussed in standard differential equations literature. We will revise the text to clarify that no formal derivation is provided, and instead cite the relevant structural property of rectified flows.
>
> **A6: Regarding the comment on diffusion models and DDIM .** We agree that deterministic sampling methods like DDIM are well-known and valid. Our statement in the appendix was not meant to suggest that all diffusion models are inherently stochastic, but to explain why we chose rectified flow models: they offer deterministic forward integration with a guaranteed likelihood formulation, which aligns well with our need for stable, continuous interpolation in latent space.
>
> **A6: Regarding the Figure 7 layout .** We agree that the caption and layout of Figure 7 could be clearer. Each row individually represents a separate morphing process with different interpolation thresholds, and the leftmost column remains fixed as the source shape. The results along the vertical axis (column) are independent. The far-left column doesn't exhibit texture changes because they are all the same 3D sources generated from the input source image. We will revise the caption and figure layout to make this clearer and indicate that the figure should be viewed horizontally, and clarify the relationship between the rows and the varying thresholds.
>
> **Weakness 3: Significance.**
>
> **A7:** While our approach builds on established components (e.g., flow-based generation and optimal transport), it is far from a simple combination of existing tools. Instead, it is a carefully designed framework that specifically addresses the unique challenges of textured 3D morphing.
>
> * The **sequential initialization strategy** serves as a **trajectory memory mechanism** in high-dimensional latent spaces, which helps stabilize barycentric optimization and avoids abrupt geometric artifacts—a failure commonly observed in naïve interpolation (see Fig. 6 and Fig. 8). Without this step, as shown in Appendix Table A1, visual quality significantly declines, highlighting its importance in ensuring smooth transitions.
> * The similarity-guided **selective interpolation** is not a trivial heuristic, but a principled mechanism that enables localized control over texture blending, which is critical for **semantic consistency and identity preservation** across the morphing sequence. Removing this step, as shown in the “Ablation study on texture interpolation strategy”, leads to blurry and incoherent textures, diminishing overall visual fidelity.
>
> **Weakness 4: Originality.**
>
> **A8:** We appreciate the reviewer’s comment and agree that optimal transport (OT) has been explored in prior works on 3D shape analysis and morphing. However, we would like to clarify that the core novelty of our work does not lie in using OT per se, but in how we formulate and apply free-support Wasserstein barycenter optimization within a flow-based generative framework for textured 3D morphing.
> To the best of our knowledge, we are the first to leverage OT barycenters in the latent token space of a conditional 3D generator, enabling:
> * Semantically meaningful, geometry-aware interpolation in a high-dimensional flow space
> * Fine-grained texture control through selective semantic blending—something not addressed in prior OT-based morphing methods
>
> In contrast, previous works typically：
> * Apply OT to fixed geometric representations, which do not incorporate semantic texture control or handle arbitrary image and text inputs in a unified framework.
> * Do not operate on latent token spaces with full texture control, limiting their ability to handle complex texture details in high-dimensional latent spaces.

---

> > ### Comment · Reviewer_pJqM · 2025-08-04
> > **Response to authors rebuttal**
> >
> > I am grateful to the authors for their detailed rebuttal.
> >
> > The authors have provided new quantitative results which address my
> > main concerns - showing that the improvement is not solely due to
> > using Trellis, and validating their method's
> > selective interpolation strategy. I also appreciate their promise
> > to revise the text and figures, to improve the clarity of the manuscript.
> >
> > Although I still believe that the proposed method's
> > novelty is limited, I am willing to raise my score.

---

> > > ### Author Response · Authors · 2025-08-05
> > > **Response to Reviewer Comment**
> > >
> > > We sincerely thank the reviewer for thoughtful feedback and for taking the time to engage with our rebuttal in detail. We truly appreciate the reviewer's acknowledgment of our new experimental results and your willingness to raise the score. We will make sure to reflect all promised clarifications and revisions in the final version. The comments have been very helpful in improving the clarity and completeness of our work.

---

### Official Review · Reviewer_qxwi · 2025-07-03

**Clarity:** 3
**Significance:** 2
**Originality:** 2
**Rating:** 4
**Confidence:** 4

**Summary:**

The paper proposes Wukong's 72 Transformations, a training-free framework for high-fidelity textured 3D morphing using flow-based generative models. By formulating shape interpolation as a Wasserstein barycenter problem and introducing a recursive initialization strategy, it achieves smooth geometric transitions. Additionally, a similarity-guided texture interpolation mechanism preserves fine texture details. Wukong's 72 Transformations supports both image and text prompts and outperforms existing 2D and 3D morphing methods in fidelity, smoothness, and semantic consistency.

**Questions:**

see weaknesses

**Ethical Concerns:**

["NO or VERY MINOR ethics concerns only"]

**Final Justification:**

The new experiments on the Headspace dataset help demonstrate real-world applicability, and the efficiency comparison clarifies that the method is competitive in inference time. The clarifications on the modular, training-free design also highlight its flexibility and future-proof potential.
While some components are adapted from prior works, the formulation of 3D morphing as a Wasserstein barycenter problem, the sequential initialization scheme, and the selective texture interpolation are non-trivial contributions that strengthen the methodological novelty. Although I still see room for broader real-world evaluation, the rebuttal has addressed my main concerns to a large extent.

Given these improvements and clarifications, I increase my score.

**Limitations:**

yes

**Paper Formatting Concerns:**

The reference style does not appear to align with the official NeurIPS template.

**Quality:**

3

**Strengths And Weaknesses:**

Strengths
1.The paper introduces a novel, training-free framework Wukong's 72 Transformations for textured 3D morphing using optimal transport and flow models.
2.It achieves high-quality results with smooth shape transitions and detailed texture preservation.
3.The method supports both image and text inputs and generalizes well across object categories.
Weaknesses
1.While the training-free nature of WUKONG is a major strength in terms of usability and scalability, it also introduces limitations. Specifically, the method depends entirely on the capacity and priors of existing pretrained models (e.g., Trellis, CLIP, DINOv2). This reliance restricts adaptability: if the pretrained model struggles with certain categories or domains (e.g., noisy real-world scans or rare object classes), WUKONG has no learning mechanism to correct or fine-tune behavior. In contrast, learning-based methods—although more resource-intensive—can be adapted to specific datasets or tasks via supervised fine-tuning.

2.While the use of optimal transport and flow models for 3D morphing is well-motivated and technically elegant, it mainly adapts existing tools rather than introducing fundamentally new modeling paradigms. The formulation of morphing as a Wasserstein barycenter problem is novel in this application domain, but the components (e.g., barycenter optimization, flow-based generation) are largely drawn from prior works. As a result, the paper’s novelty lies more in integration and application than in methodological invention.
3.Experiments focus on curated or synthetic data; real-world performance is not fully explored.
4.Inference time (30s per sample on A100) may limit practical scalability. It is suggested to compare with other methods on the inference speed and computation burden.

---

> ### Author Rebuttal · Authors · 2025-07-30
>
> We thank the reviewer for the constructive comments. We are pleased that our method's ability to achieve smooth shape transitions and detailed texture preservation, as well as its generalization across object categories with both image and text inputs, was recognized. We address the concerns below.
>
> **Concern 1: While the training-free nature of WUKONG is a major strength in terms of usability and scalability, it also introduces limitations. Specifically, the method depends entirely on the capacity and priors of existing pretrained models (e.g., Trellis, CLIP, DINOv2). This reliance restricts adaptability. & Experiments focus on curated or synthetic data; real-world performance is not fully explored.**
>
> **A1:** We appreciate the reviewer’s thoughtful analysis. We would like to emphasize that our framework is intentionally modular and training-free by design, which brings several unique strengths and flexibility:
>
> 1. **Future-proof and upgradable via backbone improvements:** Our method can immediately benefit from advances in foundational models. For example, replacing Trellis with future more powerful generators can directly improve morphing quality without retraining. This flexibility also allows adaptation to models pretrained on different domains. For instance, as shown in Appendix Table A1, we applied our morphing pipeline on top of TripoSG without other modifications, and the results are still the best among all other methods. This demonstrates the plug-and-play nature of our framework and its ability to inherit capabilities from better 3D priors as they emerge.
>
> 2. **Challenges for training-based 3D morphing methods:** We note that building a learning-based 3D morphing system in the current landscape faces significant practical challenges. Specifically, there is no readily available dataset containing sufficient 3D morphing pairs—that is, sequences of high-quality 3D models capturing smooth transitions between two given concepts, with consistent geometry and texture supervision across frames. Constructing such a dataset is particularly difficult in open-vocabulary or cross-domain scenarios (e.g., “cat → dragon” or “human → cartoon Ironman”), where intermediate forms are ambiguous or subjective.
>
>     In contrast, WUKONG’s training-free design provides a flexible, data-efficient alternative. It enables 3D morphing between arbitrary inputs (text or image) using only pretrained generative and perceptual models, without relying on morphing trajectories as supervision. This makes it immediately applicable to a wide range of prompt combinations, including those beyond curated or canonical categories.
>
> 3. **Generalization to real 3D data:** To evaluate the method’s generalization to real 3D data, we conducted experiments using the Headspace dataset[1], which contains high-quality 3D face scans along with corresponding rendered RGB images. In our pipeline, we used these rendered images as inputs and passed them through the DINOv2 encoder to extract texture and semantic features for morphing. The outputs were generated by the standard WUKONG pipeline without any architectural changes or fine-tuning. Despite relying on pretrained components, our method shows strong generalization to real-world 3D scans. Our method outperformed both MorphFlow and 3DRM(Our own implementation) on the same evaluation protocol. Quantitative results are shown below:
>
> **Quantitative results on Headspace dataset**
> | Model     | FID↓ |  STP-GPT↑ | SEP-GPT↑ | PPL↓ | V-CLIP↑
> | ----------- | ----------- | ----------- | ----------- | ----------- | ----------- |
> | MorphFlow      | 95.24   | 0.53| 0. 47 | 3.22 |  0.84 |
> | 3DRM     | 6.61   | 0.83 | 0. 77 | 3.04 |  0.88 |
> | Ours     | **3.97**   | **1.00** | **1.00** |  **2.88** | **0.96** |
>
>  We will clarify and expand these examples in the revision to emphasize real-world applicability.
>
> [1] Pears, N. E. (Creator), Duncan, C. (Creator), Smith, W. A. P. (Contributor), Dai, H. (Contributor) (6 Jun 2018). The Headspace dataset. University of York. 10.15124/6efa9588-b715-44ec-b7bb-f10dff7ca93e
>
> **Concern 2: the components (e.g., barycenter optimization, flow-based generation) are largely drawn from prior works. As a result, the paper’s novelty lies more in integration and application than in methodological invention.**
>
> **A2:** While the building blocks are rooted in existing methods, their composition and adaptation in this context form a methodologically novel and non-trivial contribution that we believe will inspire further work in learning-free 3D content manipulation.
>
> 1. We are, to our knowledge, the first to formulate 3D morphing as a free-support Wasserstein barycenter problem, enabling semantically meaningful, geometry-aware transitions in a continuous latent space.
>
> 2. We introduce a sequential initialization scheme that is critical for ensuring stable interpolation in high-dimensional flow latent spaces, addressing issues not tackled by prior work in either 2D morphing or general OT literature.
>
> 3. We further propose a similarity-guided selective texture interpolation mechanism, offering controllable and detail-preserving transitions in 3D appearance, going beyond naïve interpolation schemes used in prior art. 1.1-1.3 one point.
>
> **Concern 3:Inference time (30s per sample on A100) may limit practical scalability. It is suggested to compare with other methods on the inference speed and computation burden.**
>
> **A3:** We acknowledge that we did not include inference time comparisons in the main paper and will add this comparison in the revised edition. We report the time consumption in the following table. Our method is the most efficient one among all methods.
>
> | Model     | Morphflow |  3DRM | Ours |
> | ----------- | ----------- | ----------- | ----------- |
> | Time | 30 mins | 15 mins | 30 secs |

---

> > ### Comment · Reviewer_qxwi · 2025-08-07
> >
> > I appreciate the additional evidence provided. The new experiments on the Headspace dataset help demonstrate real-world applicability, and the efficiency comparison clarifies that the method is competitive in inference time. The clarifications on the modular, training-free design also highlight its flexibility and future-proof potential. Given these improvements and clarifications, I am willing to increase my score.

---

> > > ### Author Response · Authors · 2025-08-07
> > >
> > > We thank the reviewer for the valuable feedback and for recognizing the new evidence we added during the rebuttal. We are pleased that the experiments on the Headspace dataset and the efficiency comparison have helped clarify the versatility and practical performance of our approach.
> > >
> > > The decision to increase the score is greatly appreciated, and the comments received will be carefully reflected upon in preparing the final version of the paper. Sincere thanks again for the thoughtful review and for supporting the revisions made.

---

### Official Review · Reviewer_43Xk · 2025-07-03

**Clarity:** 4
**Significance:** 3
**Originality:** 3
**Rating:** 5
**Confidence:** 4

**Summary:**

This work proposes WUKONG, a training-free method for joint shape and texture morphing between two 3D assets. Building on top of the latent space of the TRELLIS (CVPR 2025), the authors propose an optimal-transport-based latent interpolation method to interpolate between shape latent space and the texture latent space. The proposed free-support Wasserstein barycenter optimization method takes the source and target image prompt feature tokens and builds an emperical distribution (using Dirac delta function with uniform weights), then optimizes the interpolated tokens (support points in the latent space) to minimize the Wasserstein barycenter distance. The authors solve this for shape space interpolation, then find the texture latent code, using the semantic consistency evaluation. The 3D generation pipeline itself borrows the pre-trained TRELLIS model. The authors use several plausibility and perceptual score metrics, including GPT scores, FID and PPL, etc.

**Questions:**

No outstanding questions (already discussed in weakness section).
It'd be good if the authors can also showcase the results for real images or real assets.

**Ethical Concerns:**

["NO or VERY MINOR ethics concerns only"]

**Final Justification:**

The answers mostly address this reviewer's concerns. This reviewer believes that the combination of existing concepts to build a working solution for an interesting problem can also be viewed as a novelty. Still, this reviewer requires the authors to include some real scan examples and a discussion of T-pose artifacts in the final draft. Thus, this reviewer keeps the rating.

**Limitations:**

yes

**Quality:**

4

**Strengths And Weaknesses:**

## Strengths
- The problem formulation is interesting, and the solution is simple yet effective. While the authors rely on the pre-trained powerful 3D generation model TRELLIS, the method to enable smooth shape & texture interpolation from the learned latent space is inspiring and novel. Moreover, extensive evaluations and visualizations support that the naive linear interpolation of the latent codes are a sub-optimal, but the proposed optimal-transport barycenter method is effective.
- The proposed tricks (recursive support initialization for Wasserstein barycenter problem and selective interpolation) are technically sound and effective.
- Qualitative results look promising.



## Weaknesses
This reviewer could not find significant issues or weaknesses. A few questions:
- The method seems to work on some fictional images or artistic prompts. If one wants to use it for real 3D data, e.g., human face scans, which modifications can be applied? For example, this reviewer wants to know what happens if one wants to interpolate between real human face image and artistic human face image (e.g., ironman).
- What could be the reason for T-pose human artifacts when using linear interpolation (Fig. 6, first row)? Any insights on this issues?

---

> ### Author Rebuttal · Authors · 2025-07-30
>
> We sincerely thank the reviewer for their thoughtful and positive feedback. We are pleased that the novelty and effectiveness of our approach, particularly in enabling smooth shape and texture interpolation, was recognized. We address the concerns below.
>
> **Concern 1: The method seems to work on some fictional images or artistic prompts. If one wants to use it for real 3D data, what would happen, and what modifications can be applied?**
>
> **A1.1: Current model's ability for real 3D data.**
>
> To validate the current model's ability for real 3D data, we conducted extensive tests on **real human face scans** from the Headspace dataset [1], as well as **cross-domain conditions** like real photo → stylized identity (e.g., human face → Ironman). Our method handled these scenarios successfully.
>
> 1. **Experiments between two real human faces (Headspace dataset [1]):** To evaluate the method’s generalization to real 3D data, we conducted experiments using human face renderings from the Headspace dataset, which contains high-quality 3D face scans along with corresponding rendered RGB images.
>
>     Despite the domain gap between real scanned faces and the Trellis training distribution, our method produced smooth and coherent 3D morphing sequences between different real identities. Quantitatively, using the same evaluation metrics as in Sec. 4.1.1, we obtained: **FID: 3.97, PPL: 2.88, V-CLIP: 0.91** (see "Quantitative results on Headspace dataset"). These results are on par with our main test set results (main paper table 1), suggesting that our method generalizes well to real 3D data, even under a domain shift and without any additional learning. However, despite the good quantitative results, we still observe minor qualitative degradation for the fine-grained geometric details (e.g., wrinkles or subtle facial curvature). This is most likely due to the domain bias inherited from the pre-trained trellis model.
>
> 2. **Experiments between a real human face and an artistic target:** We also tested morphing between a real human face image (photo) and a stylized image of Ironman’s helmet. The resulting 3D sequence showed a smooth and visually consistent transition, with clear semantic interpolation from the human facial structure to metallic plating, visor formation, and color/style blending. The quality is visually comparable to our result in Figure 4 (last row: "bust of Caesar" → "Darth Vader helmet"). Due to the limited number of such stylized test cases, we do not provide a quantitative average, but the results we obtained were qualitatively compelling and robust, reinforcing the method’s flexibility in handling cross-domain, real-to-stylized scenarios.
>
> Note that we conducted these experiments without any architectural changes or further fine-tuning. We will include these additional qualitative results and metrics in the updated version of the paper to better illustrate the method’s effectiveness under real-to-stylized morphing and real 3D scan domains.
>
> 3. Our current implementation is based on Trellis, which is trained mainly on synthetic textured 3D objects. Therefore, it's reasonable to perform best when the input domain aligns with the training distribution. However, we want to highlight that our method (WUKONG) is a modular, training-free morphing framework, not tied to any specific backbone. During application, it can be easily adapted to more generalizable or domain-specific 3D generators without retraining.
>
> **A1.2: Modifications can be applied.**  To further improve performance on real 3D data, the following modifications can be tried:
> * Model adaptation:  Fine-tune the encoder/decoder on real 3D dataset such as human face scans.
> * Incorporating 3D scan priors (e.g., via FLAME or DECA) to regularize the geometry flow in our framework.
> * If real 3D scans are available, the source can be used directly in Trellis’s latent geometry representation to avoid re-encoding from image.
>
> **Concern 2: What could be the reason for T-pose human artifacts when using linear interpolation (Fig. 6, first row)? Any insights on these issues?**
>
> **A2:** In flow-based 3D generators like Trellis, condition tokens (e.g., from CLIP/DINOv2) guide the generation by modulating cross-attention at each layer. Linear interpolation between latent features explores intermediate regions of the latent space. However, when the source and target features are semantically distant (e.g., from “T-Rex” to “cat”) and lack appropriate supervision, the model's conditioning may become ambiguous or collapse into mode-averaged representations. This often results in "hallucinated" generic outputs, such as T-poses or default humanoid templates commonly seen in generative models.
>
> This issue highlights the importance of our **barycentric interpolation + sequential initialization** strategy, which ensures that: 1. intermediate tokens remain on a semantically valid manifold, 2. abrupt jumps across unrelated modes are avoided. We will further clarify this insight in the revised paper and illustrate it with additional examples.
>
> [1] Pears, N. E. (Creator), Duncan, C. (Creator), Smith, W. A. P. (Contributor), Dai, H. (Contributor) (6 Jun 2018). The Headspace dataset. University of York. 10.15124/6efa9588-b715-44ec-b7bb-f10dff7ca93e

---

> > ### Comment · Reviewer_43Xk · 2025-08-05
> >
> > Thanks to the authors for their rebuttals. The answers mostly address this reviewer's concerns.
> >
> > This reviewer has read the reviews from the other reviewers and the authors' response. As some of the other reviewers have pointed out, this reviewer also noticed that this work combines multiple existing concepts or models to solve the target task. However, this reviewer believes that the combination of existing concepts to build a working solution for an interesting problem can also be viewed as a novelty. Still, this reviewer requires the authors to include some real scan examples and a discussion of T-pose artifacts in the final draft.

---

> > > ### Author Response · Authors · 2025-08-05
> > > **Response to Reviewer Comment**
> > >
> > > Thanks for the thoughtful and positive feedback. The recognition of our work’s effectiveness and novelty is deeply valued. We are grateful for the acknowledgment of the contributions made by our approach.
> > >
> > > Regarding the concerns about real scan examples and T-pose artifacts, we will include additional results using the Headspace dataset to demonstrate how the method generalizes to real 3D data. This will show how the method can be applied to real scan examples, including human face data. We will also provide a more detailed discussion of the T-pose artifacts, explaining why the artifacts emerge, how our method addresses this issue, and ensure clarity on this point in the final draft.
> > >
> > > We sincerely appreciate the constructive feedback. The comments provided by the reviewer are very helpful and will guide the final revisions of the paper.

---

### Author Response · Authors · 2025-08-02
**Rebuttal Summary and Response to Reviewer Comments**

We sincerely thank all reviewers for their valuable and encouraging feedback. We are pleased that the method was recognized as **“simple yet effective”** (R1 43Xk), capable of producing **“high-quality results with smooth shape transitions and detailed texture preservation”** (R2 qxwi), and delivering **“impressive qualitative and quantitative results, especially on challenging shape transitions”** (R4 WDq7). We also appreciate the recognition of the method’s fast inference time (30 seconds) and its potential as **“a usable tool to generate 3D assets as part of a graphics pipeline”** (R3 pJqM).

Several reviewers raised common and important concerns, which we have addressed thoroughly:
* **Results by using other model as backbone generator:** To clarify that our improvements are not merely due to a stronger generator (e.g., Trellis), we re-implemented our method on **GaussianAnything**, the same backbone used in 3DRM. As shown in **Table: Quantitative comparison with GaussianAnything as backbone**, our method consistently outperforms 3DRM across all metrics, demonstrating that the core performance gains come from our morphing strategy itself.
* **Generalization to real 3D data:** We conducted new experiments on the **Headspace dataset**, which contains high-quality human 3D face scans. As shown in **Table: Quantitative results on Headspace dataset**, our method achieves strong results in this domain as well, without any retraining or architectural changes, supporting its robustness and generalizability.
* **Practicality and speed:** Our method completes morphing in **\~30 seconds**, significantly faster than comparable methods such as **3DRM (\~15 minutes)** and **MorphFlow (\~30 minutes)**. This efficiency makes WUKONG a practical tool for real-world 3D content creation and integration into graphics pipelines.
* On the question of **novelty**, we clarified that our contribution lies not in the use of OT per se, but in its formulation as a **free-support Wasserstein barycenter in flow model latent token space**, combined with **recursive initialization** and **semantic-guided selective interpolation**. These components are critical to stability and quality, as confirmed by our ablation studies, and are not found in prior 3D morphing or OT literature.

In addition, we have addressed several representative reviewer-specific concerns, such as:
* **Explanation for the T-pose artifact** seen in baseline methods and why our method avoids it (R1 43Xk);
* Discussion on the **challenges of training-based methods** for 3D morphing due to lack of morphing-pair data (R2 qxwi);
* Inclusion of **quantitative ablation results** validating our selective interpolation strategy (R3 pJqM);
* Addition of **evaluations across different interpolation thresholds** τ for texture blending (R4 WDq7);
* Clarification on the choice of **distance metric and optimization solver** used in Wasserstein barycenter computation (R4 WDq7).

We believe we have carefully and thoroughly addressed all reviewer comments and clarified the scope and significance of our contribution. We would greatly appreciate any further feedback or questions, and we warmly welcome continued discussion during the rebuttal phase.

---

### Comment · Area_Chair_5soA · 2025-08-03
**Reminder: Discussion and Final Rating Update**

Dear Reviewers,

As we are now midway through the discussion phase, I would like to kindly remind you to review the authors' rebuttal and participate in the discussion. Please also update your review with a final rating accordingly.

Thank you very much for your time and valuable contributions to the review process.

Best regards,

Area Chair

---

### Note · Authors · 2025-08-11

We sincerely thank all reviewers for the constructive feedback and engaged discussion during the review process. These exchanges have helped refine the presentation and better highlight the contributions of our work.

The discussion has clarified several key strengths of the paper: (1) when using GaussianAnything as the backbone, our method still achieves superior performance across all metrics, confirming the effectiveness of the proposed strategies beyond backbone choice; (2) its ability to generalize to real 3D data, as shown in tests on high-quality scanned assets; (3) its competitive inference efficiency; and (4) the novelty in combining Wasserstein barycenter optimization with recursive initialization and selective interpolation to achieve smooth, geometry- and texture-aware morphing in a training-free setting.

We are encouraged and grateful that our rebuttal and the constructive discussions with the reviewers have led to unanimous support for accepting our work. We are grateful to reviewer 43Xk for the strong endorsement and accept rating, and to reviewers qxwi, pJqM, and WDq7 for the positive reconsideration reflected in their raised scores. This broad support underscores the value of a practical, training-free framework for high-quality 3D morphing that can directly benefit from future advances in generative models.

We appreciate the engaged discussion with all the reviewers in improving our paper and are committed to reflecting all the discussions into our revision.

---

### Decision · Program_Chairs · 2025-09-17

**Decision:**

Accept (poster)

**Comment:**

This paper has received consistent positive ratings of 4, 4, 4, and 5. Reviewers highlighted the novelty of the problem definition, the quality of visual results, the completeness of the work, and the clarity of writing as its main strengths. Concerns raised included the method largely building on existing work and the lack of experiments on real-world data. In the rebuttal, the authors addressed these concerns by demonstrating the difficulty of the task, which convinced the reviewers. Given the consistent positive evaluations and reviewers’ satisfaction with the responses, I see no reason to overturn their acceptance recommendation. Therefore, I recommend acceptance.